**Relationship between erythema effective UV radiant exposure, total ozone, cloud**
**cover and aerosols in southern England UK**
Nezahat Hunter, Rebecca. J. Rendell, Michael P. Higlett, John B. O'Hagan and Richard G.E.
Haylock
Public Health England, Chilton, Didcot, Oxfordshire, OX11 0RQ United Kingdom
Correspondence to: Nezahat Hunter (Nezahat.Hunter@phe.gov.uk)
**Abstract**
Evidence for an underlying trend in the dependence of erythema effective UV radiant exposure
($H_{er}$) on changes in the total ozone, cloud cover and aerosol optical depth (AOD) have been
studied using solar ultraviolet radiation measurements collected over a 25 year period (1991-
2015) at Chilton in the south of England in the UK.
The monthly mean datasets of these measures corrected for underlying seasonal variation were
analysed.  When a single linear trend was fitted over the whole study period between 1991 and
2015, the analyses revealed that the long-term variability of $H_{er}$ can be best characterised in two
sub-periods (1991-2004 & 2004-2015), where the estimated linear trend was upward in the first
period (1991-2004) but downward in the second period (2004-2015).
Both cloud cover (CC) and total ozone (TO) were found to have a highly statistically significant
influence on $H_{er}$, but the influence of the AOD measure was very small. The Radiation
Amplification Factor (RAF) for the erythema action spectrum due to TO was -1.03 at constant
levels of CC over the whole study period, that is for a 1.0% increase in TO, $H_{er}$ decrease by
1.03%.  Over the first period (1991-2004), the RAF related to CC was slightly higher at 0.97
compared to that for TO at 0.79.  The proportion of the change in $H_{er}$ explained by the change in
CC (47%) was much greater than the proportion explained by changes in TO (8%).  For the
second period (2004-2015), the pattern reversed with the observed RAF related to TO being -
1.25, almost double that of CC (-0.65).  Furthermore, in this period the proportion of variation in
$H_{er}$ explained by TO variation was 33%, double that of CC at 16%, while AOD changes had a
negligible effect (1%).
When the data were examined separately for each season, for the first period (1991-2004) the
greatest effect of TO and CC on $H_{er}$ (i.e. the largest RAF value) was found during spring.  Spring
was also the season where TO and CC variation explained the greatest proportion of variability in
$H_{er}$ (82%).  In the later period (2004-2015), the RAF and greatest influence of TO and CC were
observed in winter (67%) and the AOD effect explained further 5% variability in $H_{er}$.
This study provides evidence that both the increasing trend in $H_{er}$ for 1991-2004 and the
decreasing trend in $H_{er}$ for 2004-2015 occur in response to variation in TO which exhibits a small
increasing tendency over these periods.  CC plays an important role in the increasing trend in $H_{er}$
for 1991-2004 than TO. Whereas for 2004-2015, the decreasing trend in $H_{er}$ is less associated
with changes in CC and AOD.

## 1 Introduction

Ultraviolet (UV) radiation is only a small portion of the radiation we receive from the sun, but it has become a topic of increasing concern because of the harmful health effects it can cause. Stratospheric ozone is a naturally occurring gas that filters the sun's UV radiation. It absorbs most of the shorter wavelength UV-B radiation, whereas longer wavelength UV-A radiation mostly passes through the ozone layer and reaches the ground (WMO 2014). However, in the mid-1970s it was discovered that the release of man-made chlorine-containing chemicals could cause stratospheric ozone depletion. In subsequent years temporary ozone holes appeared over the Antarctic and to lesser extent in the Arctic (Farman et al., 1985). Stratospheric ozone depletion was also detected over populated areas such as Australia, most notably in spring when the ozone layer over Antarctica is dramatically thinned (Gies et al., 2013). Since the late 1970s, the effects of ozone depletion on UV radiation have been the subject of a large number of studies published in the literature. These studies have demonstrated that the ozone level decreased up to the mid-1990s, resulting in an increase in the amount of UV radiation reaching the Earth's surface (WHO 2006). Concern was raised that in the long-term ozone depletion would result in significantly increased UVR which in turn may result in increased incidences of skin cancers, particularly melanoma. An increase in UVR may also cause other negative health impacts, such as sunburn, ocular pathologies, premature skin aging and a weakened immune system (UNEP 2010; WHO 2006; AGNIR 2002; Norval et al., 2011, Lucas et al., 2010). However, UV radiation exposure also has known benefits to health such as the production vitamin D, which promotes healthy bones and may help in the prevention of certain illnesses including heart diseases and cancers (Holick 2007, McKenzie et al., 2009, Young 2009, Epplin & Thomas 2010).

The Montreal Protocol came into effect in 1989, banning multiple substances responsible for ozone depletion. From 1997 to the mid-2000s it became apparent that a decline in total column ozone (TO) had stopped at almost all non-polar latitudes (WMO, 2007). However, the pace of the recovery is affected by changes in temperatures, circulation, and the nitrogen and hydrogen ozone-loss cycles (Waugh et al., 2009). The ozone level has remained relatively unchanged since 2000 with most studies reporting a plateau or a limited increase in total ozone (WMO 2014).

The most important factor affecting UV radiation at the earth's surface is the elevation of the sun in the sky - this causes terrestrial UVR to vary with time of day, day of the year and with geographical location (Diffey, 2002). Aside from solar elevation, the most significant factors affecting solar UV radiation are stratospheric ozone and cloud cover (CC). A number of other factors also affect UV, including aerosol optical depth (AOD), other aerosol optical parameters, albedo and changes in other trace gases; climate change influences many of these factors (Bais et al., 2011). These factors often interact with each other in a complex way making their effect on terrestrial UV hard to quantify.

The effects of TO, clouds and aerosols on surface UV irradiance have been studied widely in Europe (Román et al., 2015, De Bock et al., 2014, Zerefos et al., 2012, Fitzka et al., 2012, den Outer et al., 2010). These studies varied in their duration between 12 and 23 years and considered various empirical models, as well as different reconstructed models including neural network techniques and radiative transfer modelling combined with empirical relationships in various locations in Europe. The majority of these studies demonstrated that increased in surface UV radiation observed were attributed to changing in weather conditions and aerosol levels rather than the changes in ozone that revealed significant increase during their study periods. In contrast, the Belgian study reported the strong impact of TO, while the individual

contribution of AOD to the non-significant negative trend of erythema UV dose was very low (De
Bock et al., 2014). Nevertheless, the influence of the aerosols on UV irradiance is still not been
fully understood due to high spatial and temporal variability (WMO 2007).
In 1990, due to the concern that the depletion of the ozone layer would cause an increase in
population UV exposure and possible impacts on human health, the former National Radiological
Protection Board (NRPB - now part of Public Health England, PHE) set up monitoring stations at
three locations in the UK to measure terrestrial solar UV (HPA 2012). The Chilton monitoring site
is located in a rural area in the south-east of England at approximately 51.6° N, 1.3° W. An
analysis of annual erythema effective UV radiant exposure ($H_{er}$) at Chilton over 25 years (1991-
2015) has previously been carried out; the results revealed a statistically significant increasing
linear trend between 1991 and 1995 and a small decreasing linear trend from 1995 to 2015
(Hooke et al., 2016; Hooke et al., 2017). The analyses described in this paper are
complementary to those undertaken by Hooke et al., (2017), which use the same data but with
methodological differences as discussed later in the paper. In particular, this work focuses on
whether the long-term trend of monthly $H_{er}$ can be linked to changes in ozone and cloud cover,
the most significant atmospheric factors that affect terrestrial UV.
**2       Materials and methods**
**2.1      Erythema effective UV radiant exposure ($H_{er}$)**
Details of the methodology for UV monitoring at Chilton are presented elsewhere (Hooke et al.,
2017). A short description of materials and methods is given here, and additional analyses using
the same data are pointed out. Erythema effective UV irradiance in the wavelength range 280-
400 nm is measured by Robertson-Berger meters (RB-500 and RB-501 since 2004,
manufactured by Solar Light Co. Philadelphia, USA). Data from these sensors are sampled to
calculate 5 minute mean values. . To convert to $H_{er}$ per day, the erythema effective UV irradiance
data were summed up daily from half an hour before sunrise to half an hour after sunset under all
weather conditions (Hooke et al., 2016). The units of $H_{er}$ are defined as the amount of energy
(joules) deposited per square meter ($J\ m^{-2}$). The first full calendar year of measurements at
Chilton began in January 1991. The daily UV data considered here are the measurements for all
available days during the 25-year period from 1[st] January 1991 to 31[st] December 2015.
Measurements were unavailable for only 3% of the days over the whole study period. For the
modelling undertaken here these were considered as missing data, while in our previous study,
they were substituted with the average value for each day over the entire period (Hooke et al.,
120 2017).
The broadband detectors measuring erythema effective UV radiation were calibrated annually
using a co-located double-grating spectroradiometer. This spectroradiometer was calibrated and
traceable to national standards. The daily radiant exposure for 22 clear days during May–
October between 2003 and 2015 have been compared to the daily radiant exposure from the
double-grating spectroradiometer. Data from the broadband detectors was found to be within
10% of the spectroradiometer data on all these days (Hooke, 2017).
**2.2     Total Ozone**
The ground-based instruments, Dobson spectrophotometers, used to measure daily column
ozone were at the UK Meteorological (Met) Office observatory at Camborne in Cornwall (south
west of England, Latitude 50.2° N, 5.3° W) for the period 1979-2003. From January 2003, ozone
monitoring has been undertaken at Reading using Brewer spectrophotometers. These
instruments measure total column ozone, which includes stratospheric ozone as well as
tropospheric ozone in the atmosphere. Both the Dobson and Brewer ozone spectrophotometers
measure TO based on measurements of the intensity of direct sunlight at selected wavelengths.
Under cloudy conditions, TO can be derived by measuring the intensity of scattered light from the
zenith sky (Smedley et al. 2012). TO is measured in Dobson Units (DU).
These two time series of data from the Camborne and Reading sites can be combined into a
single continuous TO time series (Smedley et al., 2012) as they are located at similar latitude,
while the Reading site is closer to Chilton (30km to the south-east of Chilton). The combined
dataset is considered here as a surrogate for the TO data for Chilton over the whole period 1991
to 2015. Data and other information from these sites were obtained from the air quality website
(UK-AIR) of the UK Department for Environment Food & Rural Affairs (DEFRA). The details of
the instrumentation, the ground-based ozone data and the trend analysis of TO from these sites
from 1979 to 2008 were published previously (Smedley et al., 2012).
**2.3    Cloud cover**
The HadISD dataset was created by the Met Office at the Hadley Centre in the UK, which used a
sub-set of the station data held in the Integrated Surface Database (ISD) (Met Office Hadley
Centre, 2018). The HadISD dataset comprises various selected climate variables, including total
cloud cover (CC) data that were recorded in various weather stations globally, including in the
UK for 1931–2016 (Dunn et al., 2012, 2014 and  2016).
Station based CC data in the HadISD dataset are available in various locations for the whole of
the UK. The nearest point to the PHE building in Chilton for obtaining CC data is from the
Benson weather station in Oxfordshire (51.6° N, 1.10° W, 15km to the north-east of Chilton) and
this has been used as a surrogate CC value for Chilton. The CC data were calculated hourly
from this station's observations of total cloud amount in oktas (1 okta = cloud covering one eighth
of the sky = 12.5%). The hourly time series of daily CC values at Benson were obtained from the
Centre for Environmental Data Analysis (CEDA) for the period between 1991 and 2015. The
daily average cloud amount used here is based on the recordings at this station from 11am to
2pm GMT. 11am to 2pm GMT was selected because the $H_{er}$ values during this period contribute
a large proportion of daily $H_{er}$ (approximately 40%).
**2.4    Aerosol optical depth**
The AOD data were created worldwide in various locations from a ground-based from the
AErosol RObotic NETwork (AERONET) sun photometer (https://aeronet.gsfc.nasa.gov/).
Station based AOD data in the AERONET are available in various locations in the UK. We used
the data from the Chilbolton site that is close to the Chilton site (about 77km south of Chilton)
and situated in a very rural location in the southeast of England (51.1° N, 1.44° W). The data
were available at Chilbolton from 1st January 2006 to 31st December 2015, not the full period
(1991-2015). The AOD data at wavelength 500 nm that were used here were at data level
quality 1.5, which means they were cloud-screened and quality controlled.
This study did not take account of other potentially influential atmospheric factors such as other
AOD parameters, albedo and other trace gases because there were no data available for
southern England.

## 2.5 Estimating trends

Linear regression analyses were carried out to test whether the estimated slopes in the underlying $H_{er}$, TO or CC data show real long-term trends in in the UK. However, in order to assess the long-term trends in $H_{er}$, TO and CC, seasonal variations were removed from the monthly data. This was done by calculating the overall average $H_{er}$, TO and CC for each month and then subtracting each individual value from their associated average months over the 25 year period. For each data set, the percentage deviation from the average for the seasonal corrected monthly mean data was estimated.

Longer-term variations such as the Quasi-Biennial Oscillation (QBO) and the 11-year solar cycle were not been taken into account. Since the period of the QBO is approximately 2.3 years, it affects short-term variability rather than long-term trends. This fluctuation is small in comparison to the 25-year timescale being analysed in this paper (Harris et al., 2008; Den Outer et al., 2005). The 11-year solar cycle has a longer period and therefore has the potential to impact long-term trends, however its effect on erythema effective UV levels is small (den Outer, 2005; Diffey, 2002).

The trend analyses were performed using regression analysis of the monthly mean deviation of $H_{er}$, TO or CC data versus year and t-tests were then used to determine whether the slopes of the fitted trend models were significantly different from zero. The shape of trend in the time series was examined for $H_{er}$, TO and CC by fitting linear and non-linear models to determine whether the observed values generally increase (or decrease) over time. Further analyses were carried out by examining the changes in $H_{er}$, TO and CC separately for each season (winter, spring, summer and autumn).

The evidence for autocorrelation in the residuals of the regression analysis was tested using the Durbin-Watson (DW) statistic. This test is a well-known method of judging if autocorrelation could be undermining a model's inferential suitability (e.g. assessing the confidence in the predicted value of a dependent variable). The test compares the residual for time period t with the residual from time period t-1, developing a statistic that measures the significance of the correlation between these successive comparisons (Chatfield 1996). In this study, if there was evidence for autocorrelation, a non-parametric (distribution-free) test, the Mann-Kendall test (MK) was used in place of a parametric linear regression analysis, which can compensate for temporal autocorrelation and test if the slope of the estimated linear regression line differs significantly from zero (Mann 1945, Kendall 1975, Helsel and Hirsh 1992). If a significant trend was found from the MK test, the rate of change was calculated using the Sen's slope (SS) estimator from nonparametric method (Helsel and Hirsh 1992). If the results of non-parametric analyses were similar to those results obtained by linear regression, the results from non-parametric analyses are not presented.

The relationship between $H_{er}$, TO and CC was also examined using Analysis of Variance (ANOVA) to obtain information about levels of variability within a regression model and to form a basis for tests of significance. The correlation coefficient value ($r^2$) was calculated to determine a measure of the strength of the relationship between $H_{er}$, CC and ozone and to quantify how much of the total variation in $H_{er}$ could be explained by ozone or CC. A significance level $p<0.05$ was considered statistically significant.

## 3 Results

### 3.1 Erythema effective UV radiant exposure

Summary statistics for the daily $H_{er}$ are presented in Table 1. Over 25 years $H_{er}$ ranges from 10 J m$^{-2}$ (measured on 9 January 1992, 18 days after winter solstice) to 5655 J m$^{-2}$ (measured on 20 June 2003, at the summer solstice) with a mean of 1303 J m$^{-2}$.

Figure 1 shows the distribution of the daily $H_{er}$ using boxplots for each season at Chilton. Each box shows the lower 25% quartile Q1 and the upper 75% quartile Q3 with the median as the central line. The whiskers extend from Q1 to the smallest data point and from Q3 to the largest data point. The observed results show that $H_{er}$ is highest in the summer months and lowest in winter months, while during spring and autumn months, $H_{er}$ may change rapidly day to day (Fig.1). After 2007, in particular in spring and summer it appears that $H_{er}$ values are well below their expected mean values. Data falling outside whiskers are possible extreme values with the majority observed in winter and others spread over the seasons. These extreme values could be related to natural variability in factors that affect $H_{er}$, therefore, no extreme data points were excluded from this study.

**Figure 1**: Boxplots of the daily $H_{er}$ data for each season at the Chilton site between 1991 and 2015 (grey solid line represents the mean value for each season)

Figure 2a shows the monthly mean deviation of $H_{er}$ values expressed as percentages. There is a consistent rise between 1991 and 2003 with a clear peak in 2003 when the $H_{er}$ values were the largest recorded at Chilton over the 25 year period. Thereafter, $H_{er}$ values appear to decrease. Fig. 2b also shows the mean deviation data in $H_{er}$ for each of the four seasons over the 25 year period. Winter and spring exhibited greater variability in comparison with summer and autumn, although summer has the greatest $H_{er}$ overall and therefore the largest impact on annual $H_{er}$. The large variation in during the winter and spring months is likely caused by the high variability of TO. During winter months, peaks in $H_{er}$ were observed in various years; however, $H_{er}$ in winter was very low (Table 1), so the effect on annual $H_{er}$ is small. For spring months, clear peaks are observed in March 1997, April 1997 and March 2003 (Fig. 2b). For autumn months, peaks are observed in November 2006 and 2007. Overall there has been less variability over the last few years.

**Figure 2:** Monthly mean deviation of $H_{er}$ data at Chilton (1991-2015) with trend lines (a) all seasons combined (b) seasonal.

The regression analyses of $H_{er}$ data indicate that the best fitting single linear trend covering the whole period 1991-2015 has a downward slope but that this slope is not statistically significantly different from a constant value over this period (p=0.27). Two further models were also examined. One is a linear-quadratic function, (LQ), a 2$^{nd}$ degree polynomial, which allows for more gradual variation in the monthly $H_{er}$ across the 25 year period and a second model consists of two linear trends with a node to allow for a single change in linear trend over the 25 year period. Figure 2a also shows the results of fitting these four models to the monthly mean deviation of $H_{er}$ data.

The best fitting model was the last of these, which had two linear trends that describe an
increasing trend from 1991 to 2003 and a decreasing trend thereafter, defining 2003 as the node
between two trend lines.  The nodal year appeared to be influenced by the particularly high
observations in 2003 (Fig. 2a).  In order to avoid bias that might be caused by the highest $H_{er}$
values observed in 2003, the year 2004 was chosen to be the nodal point in preference to 2003.
Based on the results of the initial model fitting to the whole period statistical analyses were also
carried out to investigate the long-term variability of $H_{er}$ for two sub-periods (1991-2004 and
2004-2015).  Table 2 presents the estimated linear slopes in percentage change per year in $H_{er}$
with 95% confidence intervals (CI) and p-values of the associated significance tests.  There is
evidence of a statistically significant increasing trend for the first period (1991- 2004) with a mean
rate of 1.01% per year ($y^{-1}$) (95% CI: 0.75%; 1.27%, p<0.001) and a decreasing trend for the
second period (2004-2015) with a mean rate of 1.35% $y^{-1}$ (95% CI: 1.98%; 0.72%, p<0.001)
based on all the data.  While there was evidence of autocorrelation, the results of the non-
parametric analyses, which would not be influenced in the same way by the autocorrelation, were
similar to those in Table 2 so they are not presented here.
For seasonal trends, the only significantly increasing linear trend was seen in winter from 1991-
2004; however, $H_{er}$ in winter was very low and contributed only a small proportion of the total
annual $H_{er}$.  The highest values of $H_{er}$ observed in summer did not show any significant linear
trend for 1991-2004 as $H_{er}$ was steady for this period.  The absence of a significant trend in
spring for this period might be explained in part by the influence of the fairly stable values of $H_{er}$
seen between 1998 and 2002 (Fig.2b).  Across the same period in autumn, the trend was found
to be approaching statistical significance (p=0.07).  For 2004-2015, the estimated trend slope
was negative for each season, but the trend was only statistically significant in summer and
autumn (Table 2).
**3.2    Total ozone**
From the combined Camborne and Reading dataset covering the period 1991-2015 TO range
from a low of 177 DU (measured in January 2006 in Reading) to a high of 524 DU (measured in
February 1991 in Camborne) with an overall mean value of 327 DU.  The distribution of the daily
TO values is presented in box plots for the period 1991-2015 for each season (Fig. 3a).  The
mean value is shown with a grey dashed line and the bold line at 300 DU shows the average
amount of TO in the atmosphere (http://ozonewatch.gsfc.nasa.gov).  Both graphs show a large
spread of TO measurements.  The data appear to be varying year to year and there is a much
larger spread of TO values in winter and spring compared to summer and autumn.  Due to
natural variability of the TO data, the extreme data points are not likely to be erroneous readings
and so they were not excluded from the analysis.  In general the TO values were low in autumn
and early winter with a few exceptional cases in March, April and August.  Similarly, the
maximum TO values were mostly found in late winter and in early spring (Fig. 3b).  In Fig. 3b, the
solid black line indicates the overall mean value (327 DU) and the grey dashed line represents
the baseline ozone level of 220 DU which is the threshold used to define the Antarctic ozone hole
(https://ozonewatch.gsfc.nasa.gov).
**Figure 3**: Daily TO values: (a) Box plots for each season for the period (1991-2015) in southern
England, (b) Line plots for the period 2005-2015 at Reading.
Table 3 presents the estimates of the linear slopes in percentage change per year in total ozone
data with 95% confidence intervals (CI).  The regression analysis of the trend for the period
1991-2015 showed a highly statistically significant increasing linear trend of 0.17% $y^{-1}$ (95% CI:
0.09%; 0.25%, p<0.001).  The evidence for autocorrelation in the residuals of this regression
analysis was tested and the DW test confirmed that the overall level of autocorrelation in the
residuals was highly statistically significant (p<0.001).  Applying the non-parametric MK test to
these data also indicated a strongly statistically significant increasing trend in the TO across the
full study period and the Sen's slope median trend estimate was 0.13 % $y^{-1}$ (95% CI: 0.05%;
0.21%, p<0.001).  This slope estimate was smaller than that obtained by the linear regression
analysis (Table 3).
A model consisting of two trend lines with a node at 2004 was fitted to these data and the results
are shown in Table 3.  The regression analysis gave slightly different results to those obtained
using the non-parametric methodology.  The regression analysis found an increasing trend of
0.19% $y^{-1}$ in TO for the period 1991-2004 which was a borderline statistically significant (p=0.06)
and a statistically significant (p=0.03) upward trend for 2004-2015 with a value of 0.28% $y^{-1}$ (95%
CI: 0.003; 0.53).  The non-parametric test also showed that the slope of the trend during 1991-
2004 was positive (0.16% $y^{-1}$; 95%CI: -0.02; 0.35) but not statistically significant (p=0.09), while
in the latter period the slope trend was positive, 0.22% $y^{-1}$ (95% CI: 0.002%; 0.44%), but this
result was of borderline statistical significance (p=0.05).
The comparable analyses of the seasonal data are presented in Table 3.  The trend for the TO
data was only statistically significant in winter over the period 1991-2015 (0.43% $y^{-1}$; 95% CI:
0.19%; 0.67%, p<0.001).  While there was evidence for autocorrelation in the residuals, Sen's
slope trend estimates were found to be very similar to the slope estimates obtained by linear
regression.  The slope of the trend estimate was positive for any of the seasons in the both
periods (1991-2004 and 2004-2015) but these increases were not statistically significant. For the
summer periods, the ozone trend was not statistically significant due to very small ozone
changes and high $H_{er}$, while  the ozone trends in winter influenced the very low $H_{er}$ at that time of
year, but had little impact on overall annual $H_{er}$.

**3.3    Erythema effective UV radiant exposure and total ozone**
Further analyses were carried out to examine the relationship between $H_{er}$ and TO for the period
1991-2015.  Fig. 4 shows an inverse relationship between $H_{er}$ and TO, $H_{er}$ being high when TO is
low and vice versa.  This is evident for all seasons (Fig.4b).  The greater variability in $H_{er}$
observed in winter and spring appears to be caused by the grater variability in TO for the same
seasons (Fig.4a).  However, the effect on $H_{er}$ was negligibly small in winter and might be
significant in spring if TO events were very low.  An inverse relationship was also observed in
summer and in autumn, but not to the same extent as that seen in winter and spring. In
particular, lower TO and higher $H_{er}$ values were observed during spring 1997 and also from
December 2011 to March 2012, but the highest $H_{er}$ values recorded at Chilton over the 25 year
period were during spring 2003 and there was no significant TO reduction over the same year. In
contrast, particular fluctuation of higher TO and lower $H_{er}$ values were observed after 2007 (Fig.
4a).  In absolute terms, all these observed changes are small and implications for $H_{er}$ in winter
are small.

**Figure 4**: Relationship between monthly mean deviation of $H_{er}$ (1991-2015) and TO (1991-2015): (a) seasonal, (b) fitted linear regression line with 95% CI.

Table 4 shows the results of the regression analyses of the monthly mean deviation of $H_{er}$ against TO by season and for all seasons together for the period 1991-2015, as well as the correlation coefficient estimates ($r^2$) for each regression model. The inverse correlation between $H_{er}$ and TO was found to be strongly statistically significant ($p<0.001$) for the period 1991-2015 such that a 1% increase in TO was associated with a 1.3% decrease in $H_{er}$. This is known as the Radiation Amplification factor (RAF) for the erythema action spectrum for sunburn of human skin, i.e. RAF for TO was -1.3. This corresponded to the observed increasing trend (0.13% $y^{-1}$) in long term TO corresponding to a negative trend of -0.17% $y^{-1}$ in $H_{er}$ when the TO trend was multiplied by the changes in $H_{er}$ (-1.3). The scatterplot between $H_{er}$ and ozone in Figure 4b also shows this slope of the trend line in which there is a wide spread of data points around the line. This indicates a weak correlation as confirmed by an $r^2$ value of 25%. However, 75% of the variation could not be explained by TO alone so other factors such as CC and aerosols are likely to be important.

The results for the two-sub periods 1991-2004 and 2004-2015 are also presented in Table 4. A statistically significant negative correlation was found between $H_{er}$ and TO ($p<0.001$) for both periods. The estimated slope was negative for both periods, where a 1% increase in TO leads to a 1.2% and 1.5% decrease in $H_{er}$ respectively. These RAF values are slightly different to those for the full study period (1991-2015), but a test of heterogeneity in the three RAF values showed that there was no statistically significant difference between them ($p=0.68$). When the ozone trend was multiplied by the RAF, the increasing ozone trend for both periods (0.16% $y^{-1}$ and 0.22% $y^{-1}$) corresponded to a negative trend of -0.19% $y^{-1}$ and -0.33 $y^{-1}$ in $H_{er}$. Only a small proportion of the variation in $H_{er}$ over the first and second period could be explained by TO (18% and 33% respectively), indicating that other factors such as CC and AOD may have a stronger influence over these periods.

For the seasonal data, the inverse correlation between $H_{er}$ and TO was also highly statistically significant ($p<0.001$) for 1991-2015 (Table 4). Variability in $H_{er}$ explained by TO rose in spring (41%) and summer (34%). A 1% increase in TO during spring and summer seasons leads to an average of 2.4% and 1.9% decrease in $H_{er}$ respectively. The RAF value in winter was less negative than the values in summer, spring and autumn. This pattern is expected since an increase in cloudiness tends to reduce $H_{er}$. A test of heterogeneity in RAF values between seasons showed no statistically significant difference in the RAF values except between winter and summer ($p<0.001$) and winter and spring ($p<0.001$).

Across both sub periods, the inverse correlation between $H_{er}$ and TO for the period 1991-2004 was statistically significant for all seasons except during winter ($p=0.12$) and the variability in $H_{er}$ explained by TO in summer was 42% and in spring was 37%. In contrast, for the period 2004-2015, variability in $H_{er}$ explained by TO was stronger in winter (52%) and in spring (48%) than in summer (24%) and autumn (17%), although $H_{er}$ in winter is very low. The test of heterogeneity in RAF values between spring, summer and autumn showed no statistically significant difference in the RAF values for the period 1991-2004 ($p>0.20$). For the period 2004-2015, there were no differences in RAF values between any of the seasons ($p=0.53$).

## 3.4    Erythema effective radiant exposure and cloud cover

The long-term changes in $H_{er}$ in all weather conditions also differ according to variations in CC.
The regression analysis of all the CC data showed a statistically significant downward linear
trend with a mean rate of 0.19% $y^{-1}$ (95% CI: -0.34%; -0.04%, p=0.01). When the data for each
season was considered separately, a statistically significant downward linear trend was only
found in spring (p=0.025) although the trend slope was negative for the other three seasons.
The regression analysis of CC for the first period (1991-2004) also showed a statistically
significant downward linear trend of 0.68% $y^{-1}$ (95% CI:-1.03; -0.33, p=0.0002), but for 2004-2015
the downward linear trend was small (-0.04% $y^{-1}$) and not statistically significant (p=0.85).
Seasonally, the slope estimates were negative for all four seasons for 1991-2004, but only the
trends for winter and spring were statistically significant (-0.93% $y^{-1}$, p=0.02 and -0.72% $y^{-1}$,
p=0.03 respectively). In contrast, for 2004-2015, there was no evidence of a trend in CC for any
season, although the trend estimate was negative for winter and spring, but positive for summer
and autumn.
Fig.5 shows the relationship between CC and $H_{er}$ for the period 1991-2015. As expected an
inverse relationship was observed and peak $H_{er}$ was seen to increase in response to decreasing
CC for all seasons.
**Figure 5**: Relationship between the mean deviation of $H_{er}$ (%) and CC at Chilton (1991-2015): (a)
seasonal, (b) correlation plot showing the linear regression line with 95% CI.
Table 5 shows the results of the regression analyses of the monthly mean deviation of $H_{er}$ data
against CC. A highly statistically significant inverse correlation was found for each season and
for all seasons together. For the whole data over the period 1991-2015 the analysis shows a 1%
increase in cloud was associated with a decrease of about 1% in $H_{er}$, i.e. RAF for CC is -1. This
slope of the trend line on a scatterplot in Fig.5b indicates modest correlation between TO and
$H_{er}$, confirmed by an $r^2$ value of 38%, leaving 62% of the variation unexplained. Seasonally,
changes in $H_{er}$ explained by CC was the highest in spring (48%) and summer (46%) over the
period 1991-2015. The decrease in the long term CC (-0.19% $y^{-1}$) corresponded to a positive
trend of 0.18% $y^{-1}$ in $H_{er}$ when the CC trend was multiplied by the RAF for CC (-1).
A statistically significant negative correlation was found for the two-sub periods 1991-2004 and
2004-2015 (p<0.001 for both periods), results presented in Table 5. While the slopes of the trend
estimates were similar for both periods (RAF for CC, -1.06% and -0.82% respectively), the
strength of the correlation between $H_{er}$ and ozone was moderate (48%) for 1991-2004, but low
for 2004-2015 (27%). The decrease in CC trend for the period 1991-2004 (-0.68% $y^{-1}$) and for
2004-2015 (-0.04% $y^{-1}$) corresponded to a positive trend of 0.72% $y^{-1}$ and 0.03% $y^{-1}$ in $H_{er}$
respectively when the CC trend was multiplied by RAF for CC -1.06% and -0.82%.
For the seasonal data for the periods 1991-2004 and 2004-2015, all slope estimates were
negative and statistically significant (Table 5). The correlation was strongest in spring (66%) and
summer (64%) for 1991-2004, but moderate for the same seasons for 2004-2015.
**3.5    Erythema effective radiant exposure and aerosol optical depth**
The long-term changes in $H_{er}$ also differ in relation to aerosols. Aerosols can affect ground level
UV irradiances directly through solar radiation absorption and scatter, reducing the amount of
solar radiation reaching the surface of the Earth. Aerosols can also affect UV levels indirectly by
modifying cloud formation (REF). Atmospheric aerosols originate from both natural sources
(such as soil dust) and from anthropogenic sources – such as air pollution from industry and
traffic in urban areas (REF).
Since both the Chilbolton and Chilton sites are in a very rural location, AOD levels at both sites
should be comparable.  The AOD showed a statistically significantly decreasing linear trend with
a mean rate of about 4.3% $y^{-1}$ (95% CI: -6.20; -2.40 $y^{-1}$, p<0.001) over the period (2006-2015).
Seasonally, a statistically significant downward linear trend was only found in winter and spring
with a mean rate of 4.6% $y^{-1}$ (p=0.003) and 5.8% $y^{-1}$ (p=0.02) respectively. The slope of the trend
for the other two seasons was also negative, but not statistically significant (p<10).
When $H_{er}$ was compared to AOD for the period 2006-2015, the estimated slope was small and
positive (0.09%) and the correlation between $H_{er}$ and AOD was statistically significant (p=0.03).
The AOD effect explained only 4% of the variability in $H_{er}$.  Based on seasonal specific analysis,
the statistically significant correlation between $H_{er}$ and AOD was found only in summer (p=0.03)
and the estimated slope was positive (0.15%), while the variability in $H_{er}$ explained by AOD was
16%.  Although the correlation between $H_{er}$ and AOD was not statistically significant for the other
three seasons, the AOD effect was the largest and positive for winter (0.21%, the smallest in the
summer  (0.09%) and  the effect was negative in autumn (-0.04%).
**3.6      Erythema effective radiant exposure, total ozone, cloud cover and aerosol**
**optical depth**
A multiple linear regression analysis was used to investigate how the variation in CC and TO
consider together relates to changes in $H_{er}$ for 1991-2015.  The results showed that CC and TO
have the largest and statistically significant influence on $H_{er}$ (p<0.001) and the results presented
in table 6.  The estimated slopes for both CC and TO were negative and statistically significant
for all seasons.  On average, $H_{er}$ decreased by 0.82% for a 1% increase in CC at constant levels
of TO.  Similarly, $H_{er}$ decreased by 1.03% for a 1% increase in TO, at constant levels of CC. The
RAF due to TO (-1.03) was slightly different to the one based on the previous model with no CC
effect adjustment, but the difference was not statistically significant (p=0.50).  The combined
model using TO and CC explained only 51% of the variability in $H_{er}$ with the individual
contributions of 14% (TO) and 37% (CC), leaving 49% unexplained.
Across the season-specific analyses, $H_{er}$ changes induced by TO and CC variability were highest
in spring (68%) and summer (55%) for 1991-2015. The individual contribution of CC accounted
for the largest variation in $H_{er}$ (47% and 46% respectively), while the variation in $H_{er}$ explained by
TO was low (21% and 12%) when CC was in the model (Table 6).  Winter changes in $H_{er}$
induced by these two factors were the lowest (42%), individual contributions were 27% (CC) and
15% (TO). For autumn the changes were moderate (51%) with variation in CC accounting for
41% and TO explaining only 10% of the total variation in $H_{er}$.   There was no statistically
significant difference in RAF values due to TO between the seasons (p>0.50), except for winter
and spring p=0.02 at constant levels of CC.
Table 7 shows the results from the multiple linear regression analysis for 1991-2004 and 2004-
2015.  For 1991-2004 at constant levels of CC, a 1% increase in TO caused $H_{er}$ to decrease by
0.79% (RAF for TO was -0.79). For 1991-2004, 55% of the total variation in $H_{er}$ was explained by
CC and TO together with respective contributions of 47% and 8%. In this case the relative
contribution from TO was small.

In contrast, for 2004-2015, of the total 49% of the variation in $H_{er}$ was explained by TO (33%) and
CC (16%). $H_{er}$ decreased by 0.65% for a 1% increase in CC at constant levels of TO, while
1991-2015 showed $H_{er}$ decreased by 0.97% for the same conditions. The RAF value for TO (-
1.25) in 2004-2015 was different to that in 1991-2004 and for 1991-2015, but a test of
heterogeneity showed that this was not statistically significant (p=0.44).
The season specific results showed that the highest correlation for the period 1991-2004 was
observed in spring (82%) and of this total 65% was explained by CC and 17% by TO. In
contrast, the highest correlation value for 2004-2015 was found in winter (67%), with 15%
explained by CC and 52% by TO. For summer and autumn, CC was found to be the larger
influence to the variation in $H_{er}$ (31% and 26%, respectively) in comparison with TO (10% and
11%, respectively), which shows a clear swap in the main contributing factor between autumn
and winter. The heterogeneity test for RAF values for TO between seasons showed no
statistically significant differences for either period (p>0.10).
When the AOD effect added to the model which also contained the TO and CC effects on $H_{er}$ for
the period 2006-2015, the estimated slope for AOD was small and positive (0.02%), but the AOD
did not lead to a statistically significantly change in $H_{er}$ (p=0.47) and contributed only 1% changes
in $H_{er}$ in comparison to the contribution from TO (40%) and CC (12%). AOD therefore has a
negligible impact.
Seasonally, the changes in AOD showed a statistically significant effect on $H_{er}$ only in winter
(p=0.04), when the AOD effect added to the model that contained the TO and CC effects for the
period 2006-2015. The total variation explained in $H_{er}$ by these three factors together was 76%
and of this total TO, CC and AOD contributed to the changes were 57%, 14% and 5%
respectively. The estimated slope for TO, CC and AOD was -1.44, -1.04 and +0.18. The AOD
effect was very small and positive in spring and summer, and negative in autumn, but the AOD
did not lead to statistically significant changes in $H_{er}$ for three seasons.
**4      Summary and Discussion**
**4.1      Erythema effective UV radiant exposure**
This paper reports an analysis of the effect of TO and CC on $H_{er}$ at Chilton between 1991 and
2015. During this period the highest values of $H_{er}$ were observed in 2003, likely due to low CC,
but not with any significant reduction in TO level. It was also the same year that a heat wave
affected much of Western Europe including England (Vieno et al., 2010; Beniston 2004).
However, hot weather does not necessarily mean high UV and cold weather does not
necessarily mean low UV (Wong et al 2015). High levels of $H_{er}$ were also reported at two sites,
Lindenberg in Germany and at Bilthoven in Holland (den Outer et.al. 2005; WMO et. al 2007) in
2003. These sites are at latitudes (49$^o$ N, 52$^o$ N respectively) which are close to that of Chilton
(52$^o$ N). Den Outer & colleagues suggested that the high annual erythema effective UV dose
received in Holland in 2003 was associated with extremely low cloud levels combined with
moderately low ozone values. However, no such associations were reported at Uccle in Belgium
with a latitude of 51$^o$ (De Bock et al., 2014) or at Reading in the UK (Smedley et al., 2012). $H_{er}$
data at Chilton also showed a reversal in trend before and after 2003 with an increasing trend
from 1991 to 2003 but a decreasing trend thereafter. In order to avoid bias in the analyses
caused by the highest $H_{er}$ values occurring in 2003, the year 2004 was chosen to be the
changing point in preference to 2003.
In our previous analysis of the long-term variability of $H_{er}$ between 1991 and 2015 at Chilton, the
data were divided into two separate time series due to geophysical phenomena. The first being
from 1991 to 1995, due to the ozone turning point in the mid-1990s (WMO 2014) and the second
time series being from 1995 to 2015 (Hooke et al., 2017). In contrast, this current work
considered the entire time period of 1991-2015 and also splits the time series according to
statistical analysis. The $H_{er}$ data for 1991-2015 (based on a nonlinear model over the full period)
were statistically better described by two linear trends; the first a statistically significant
increasing linear trend value of 1.01% $y^{-1}$ (p<0.01) for 1991-2004 and the second a statistically
significant decreasing trend of 1.35% $y^{-1}$ (p<0.01) from 2004-2015. Our finding for the first period
is consistent with our earlier result for the period 1991-1995, but a higher estimate (4.4% $y^{-1}$) was
obtained due to relatively short time period, 5 years. Our findings for the second period also
agree with those of our early study for 1995-2015 but the trend estimate was slightly lower (-
0.8% $y^{-1}$).
The finding in this study for the first period (1991-2004) is in good agreement with those from
European studies that also reported significant increasing linear trends. At Lindenberg in
Germany an increasing trend of 0.77% $y^{-1}$ during 1996-2003, 0.85% $y^{-1}$ for the period 1999-2004
and 1.4% $y^{-1}$ over the period 1998-2005 was reported. The studies at Norrköping in Sweden and
also at Bilthoven in Holland both reported an increasing trend during 1996-2004 (1.2% and
0.86% $y^{-1}$ respectively) based on solar zenith angles (SZA) of 60°, but the trend was higher (1.7%
$y^{-1}$) at Bilthoven for the period 1998-2005 when the noon values of the erythema UV radiation
were used (Bais et al., 2007). The study at the Hoher Sonnblick site in Austria (Fitzka et al.,
2012) showed a significant upward trend in the erythema weighted irradiance for the period
1997-2011 with a range from 0.84%±0.52% $y^{-1}$ at 45° SZA to 1.26%±0.36% $y^{-1}$ at 65° SZA under
all weather conditions. However, a smaller and less significant result was seen at wavelengths
of 305 nm (between -0.76%± 1.13% $y^{-1}$ and 0.79%± 0.73% $y^{-1}$, depending on SZA) at Hoher
Sonnblick. In the UK, Reading also found a significant increasing linear trend (0.66% per year)
for the period from 1993 to 2008 based on the midday values of UV Index (Smedley et. al 2012).
A significant increasing trend at 325 nm (0.34% $y^{-1}$) in Europe based on averaged data from five
selected stations was reported for the period 1995-2011 (Zerefos et al. 2012).
The trend in $H_{er}$ in this study over the second period (2004-2015) at Chilton is consistent with
values derived for the averaged UV-B data over Canada, Europe and Japan that showed
statistically significant evidence of a reduction in UV-B for the period 2007-2011 with the slope
estimates ranging from -1.5% to -2% under cloudless conditions (Zerefos et al., 2012). These
authors also showed that this slowdown of upward trend in UV-B was at a turning point after
2007 for constant cloudiness. Our findings are also in good agreement with the results of
Fountoulakis et al. (2016) at Thessaloniki in Greece, where a turning point in the trends of UV
irradiance was reported as occurring in 2006; a statistically significant increasing trend of 0.71%
±0.21% $y^{-1}$ for the period 1994-2006 and an insignificant decreasing trend of 0.33% ± 0.32% $y^{-1}$
from 2006 to 2014. It appeared that there was a similar behaviour of the trend in the UV
irradiance between this UK study and the Greek study, although these countries differ
significantly in terms of climate and location (51.6°N Chilton versus 40°N Thessaloniki).
However, a recent study at Uccle in Belgium covering the period 1991-2013 (similar to that
examined in this study period) found a strongly statistically significant increasing linear trend of
0.7% $y^{-1}$ (De Bock et.al. 2014). In comparison, our results for the period 1991-2015 found a non-
significant downward trend.

When $H_{er}$ data for each season were analysed separately, a statistically significant increasing
trend was only found in winter for the first period 1991-2004, despite large inter-month variability.
However, winter only contributes a small fraction of the total annual $H_{er}$ in the UK. Much of this
significant result might be caused by the high frequency of low TO events observed in winter
(see Section 4.2). However, there was no significant linear trend in $H_{er}$ in either spring or in
summer at Chilton. The absence of a significant trend in spring for this period might be due to
higher values of TO level over the same period.
The $H_{er}$ changes for the second period 2004-2015 showed a linear downward trend for all four
seasons, but the trend was only statistically significant in summer and autumn. The results of the
current study are comparable with those of the Belgian study at the Uccle site for the period
1991-2013. The results from the Belgian study showed the largest statistically significant
increasing trend in $H_{er}$ in spring, but a negative trend in winter, albeit not statistically significant
(De Bock et. al 2014). In addition, the Austrian study at the Hoher Sonnblick site for the period
1997-2011 found that the largest and most significant linear trends were during winter and
spring.
**4.2    Erythema effective UV radiant exposure and TO**
The significantly increasing linear trend in TO of 0.13% $y^{-1}$ (p<0.001) in the south of England for
1991-2015 could be due to natural variability in TO. This result is lower than but generally in
good agreement with the significant upward trends reported in other European studies: 0.19% $y^{-1}$
at Hoher Sonnblick in Austria during 1997-2011 (Fitzka et al., 2012) and 0.26% $y^{-1}$ at Uccle in
Belgium during 1991-2013 (De Bock et. al.,2014). Our findings are also consistent with the result
for the period 1995-2011 over Canada, Europe and Japan (Zerefos et al., 2012). The Reading
study, however, using a subset of the same TO data used here reported a small and not
significant increase over the period 1993-2008, but the estimate lay within the range of trends
observed at other European stations (Smedley et al., 2012). The analysis of seasonal data here
showed the largest and most significant increasing linear trend during winter, while there was an
upward trend in other seasons it was markedly smaller and not significant. The Reading study,
however, did not show any significant trend for any season, although an increasing rate in winter
was noted for the period 1993-2008. Unlike this study, the Belgian study at the Uccle site only
found statistically significant increasing linear trends of TO in spring and summer for the period
1991-2013 (De Bock et al., 2014).
A statistically significant inverse relationship was found between TO and $H_{er}$ (p<0.001). The RAF
due to TO suggested a 1% increase in TO was associated with a 1.3% decrease in $H_{er}$ for the
period 1991-2015, in good agreement with studies from the USA (RAF = -1.1, Hall 2017) and
Spain (RAF = -1.3 to -1.4, Antón et al. 2009). However, the variability in $H_{er}$ due to the TO was
weak (25%). This is not surprising as the amount of UV radiation reaching the Earth's surface
depends on a number of factors such as CC, atmospheric aerosols, air pollution as well as other
climate factors, not solely on TO (Calbó et al., 2005).
We found an increasing tendency trend of 0.16% $y^{-1}$ (p=0.09) and 0.22% $y^{-1}$ (p=0.05) in TO for
both periods 1991-2004 and 2004-2015. The association between $H_{er}$ and TO was found to be
statistically significant for both periods; the RAF value showed that a 1% increase in TO was
associated with a -1.2% and -1.5% decrease in $H_{er}$ for the two periods respectively. The
increase in TO trend for 1991-2004 and 2004-2015 corresponded to a negative trend of -0.19%
$y^{-1}$ and -0.33 $y^{-1}$ in $H_{er}$ respectively. However, the amount of variation in $H_{er}$ explained by that of
TO was low (18% and 33% for each time period respectively). The size of the slope estimates in
TO are smaller than those reported in the study by Zerefos et al., (2012), where increasing ozone
effect on 305nm irradiances was estimated to be on the order of -4% over the period 2007-2011.
This higher estimate obtained in that study might be due to relatively short time period (5 years)
in comparison to our study period (2004-2015, 12 years) and also their result was based on the
averaged data over Canada, Europe and Japan sites. The Belgian study (De Bock et al., 2014)
also reported a greater effect of TO on the erythema UV dose (-5%) during the period 1991-2008
when measures of global solar radiation and AOD were taken into account that resulted the
bigger TO. The Reading study, however, did not find any correlation between the surface UV
radiation and TO for the period 1993-2008. The authors suggested that the majority of the
variability in UV radiation was due to changes in CC and other effects (Smedley et al., 2012).
Examining our data on a season by season basis over the whole period from 1991 to 2015, we
found a highly negative slope estimate for each season between TO and $H_{er}$. The RAF varied
from -0.94 in winter to -2.37 in summer. However, there were some differences in the amount of
$H_{er}$ variation that was explained by TO across the seasons. In spring and summer the variability
explained was moderate at 41% and 34% respectively, but in winter and autumn it was
considerably lower at 19% and 21%.
Over the first period (1991-2004) the RAF ranged from -0.54 for winter and -2.47 for summer.
The greatest impact of TO on $H_{er}$ was seen in spring (37%) and summer (42%), but for the
second period (2004-2015), the impact was bigger in winter (52%) and spring (48%) than in
summer (24%) and autumn (17%). This was mainly due to the strong inverse relationship
between $H_{er}$ and TO that was observed during spring and winter in the period 2008-2015 (63%
and 56%), while most TO values were higher and remained stable for the same period (Fig.4).

**4.3 Erythema effective UV radiant exposure and CC**
CC can have a marked impact on the amount of UV that reaches the earth's surface. An
increase in CC usually results in a reduction of UV radiation below the clouds. Whilst UV can
pass through thin and broken clouds, thick clouds tend to reflect, absorb or scatter UV radiation.
Puffy, fair-weather clouds reflect rays and can actually increase the UV radiation reaching the
earth's surface (Alados-Arboledas et al., 2003).
There was a statistically significant decrease trend of 0.19% $y^{-1}$ in CC for the period 1991-2015.
Our analysis of CC variation also showed a small significant decreasing trend of -0.68% $y^{-1}$
(p<0.001) for the first period (1991-2004), but no significant trend for the second period (2004-
2015) although the estimated slope was small and remained negative (-0.04% $y^{-1}$). Our findings
agree partly with other studies that reported a significant decrease in CC (Norris and Slingo,
2009, Eastman and Warren, 2013) and those that did not find any evidence of a decreasing trend
in CC. For example, the studies at the Hoher Sonnblick site in Austria over the period 1997-2011
(Fitzka et al., 2012) and in the study examining data from Europe, Canada and Japan for the
period 1995-2011 (Zerefos et al., 2012).
The association between $H_{er}$ and CC was also statistically significant; a 1% decreasing trend in
CC was associated with 0.95% increase in $H_{er}$ for the period 1991-2015 and the CC explained
38% of the variability in $H_{er}$. This is higher than that the variability in $H_{er}$ explained by TO alone.
The inverse correlation between $H_{er}$ and CC was also strongly statistically significant for both
1991-2004 and 2004-2015 (-1.06% and -0.82% respectively). The decrease in CC trend for the
same periods corresponded to a positive trend of 0.72% $y^{-1}$ and 0.03% $y^{-1}$ in $H_{er}$. However, while
about half the variation in $H_{er}$ was explained by CC in the first period this fell to just over one
quarter for the second period.
Examining our data on a season-by-season basis, the only statistically significant trends in cloud
reduction observed were in spring and winter during the period 1991-2004. For 2004-2015, the
variation in $H_{er}$ explained by CC dropped below 50% for all seasons, particularly autumn (26%)
which was lower than that for winter (29%). These observations agree with the findings from the
Austria study at Hoher Sonnblick for 1997-2011 (Fitzka et al., 2012). Even though the study did
not look at the correlation between CC and UV, the authors reported that the total cloud
reduction of 1.04% $y^{-1}$ was evident for UV at SZA 55$^{o}$ for the period 1997-2011.
**4.4 Erythema effective UV radiant exposure and AOD**
The ground-based AOD measurements at the Chilbolton site showed a statistically significant
decreasing trend of -4.3% $y^{-1}$ (p<0.001) for 2006-2015. This finding agrees with the studies that
reported a declining trend in AOD in Europe during the period 2003-2015, attributed to increasing
air quality due to environmental regulations (Provençal et al., 2017, Alpert et al. 2012). Our trend
estimate is consistent with that AOD estimate derived at five stations in Europe (-4.3 $y^{-1}$), which
used the dataset from the satellite MODIS for the period 1995-2011 (Zerefos at al., 2012).
However, it disagrees with the insignificant negative trend that was found in the Belgium study at
Uccle with a slower rate of -0.8% $y^{-1}$ for the period 1991-2013 (De Bock et al., 2014). The
Austrian study at Sonnblick found even smaller decline in AOD trend of -0.5 to -0.6% $y^{-1}$ over
1997-2011 and a statistically significant and (Fitzka et al. 2012).
Seasonally, a statistically significant downward AOD linear trend was only found in winter and
spring in this study, whereas the Belgium study reported the AOD trend was negative and
significant during summer and autumn (De Bock et al., 2014).
When $H_{er}$ was compared to AOD for the period 2006-2015 in this study, the slope estimate for
AOD exhibited a small and positive (0.09%), yet significant effect on $H_{er}$ (p=0.03). Our finding
agrees with the study by Zerefos et al., (2012), but the AOD effect at 305nm irradiances -based
on data averaged over Canada, Europe and Japan sites- was bigger with a positive estimate of
a 1.8% increase over the period 2007-2011. The authors also showed that $H_{er}$ increased with
increasing AOD based on 2$^{nd}$ deg. polynomial fit as the effect of "brightening" at 305nm as a
result of reduced AOD.
Similar positive AOD effect on $H_{er}$ was also seen for the seasonal specific analysis where $H_{er}$
level increased with increasing the AOD level (except autumn) and it was only statistically
significant in summer (p=0.03). Similarly, the study at Uccle reported also positive AOD effect on
$H_{er}$ for all seasons except in summer. Authors stated that an increase in AOD could lead to an
increase in UV radiation if the increase in AOD were caused by an increase in the amount of
small scattering aerosol particles (De Bock et al., 2014). However, here we do not have any
information on these parameters.
**4.5    Erythema effective UV radiant exposure, TO, CC and AOD**
Given that we found clear evidence that variation in $H_{er}$ could be partially explained by variation
in TO and CC separately, we considered their combined effects using additive multiple linear
regression analysis. An additive multiple regression model with just TO and CC showed that for
1% change of $H_{er}$ the model factors TO and CC changed by -1.03% and -0.82% respectively for
the period 1991-2015. This RAF estimate due to TO (-1.03) is consistent with the RAF values
due to TO in the US study (Hall 2017) in which impact of clouds on the RAF was determined and
ranged from a low of -0.80 to a high of -1.38.

Over the whole period 1991-2015, half of the variation in $H_{er}$ could be explained by TO and CC
with TO accounting for 14% and CC accounting for 37% of the variation in $H_{er}$. In this work, the
combined effects of TO and CC on a season-by-season basis showed the biggest proportion of
variability in $H_{er}$ was for spring (68%), while the smallest proportion was in winter. In each
season, CC explained far more of the variability than TO from the additive linear regression
model that was used.

The combined effect of TO and CC on $H_{er}$ was assessed using two linear trends with a node at
2004. Over 1991-2004 the proportion of $H_{er}$ variability explained by these two factors rose a
small amount (4%), but fell slightly for 2004-2015 (2%) compared to the proportion for 1991-
2015. A major difference was seen in how much of the variability was explained by each factor
across the two periods. For 1991-2004, CC variation accounted for a lot more of the explained
variability compared to TO (47% in comparison to 8%), whereas for 2004-2015 the proportions
were 16% and 33% respectively. This reversal can be attributed to significant correlation
between TO and $H_{er}$ observed in winter and spring in particular during 2008-2015 which has a
bigger impact on $H_{er}$ than CC for the same period.

Across the seasons there were marked differences in the proportion of $H_{er}$ variability explained
by TO and CC. For the period 1991-2004, combining TO and CC together explained 82% of the
variability in $H_{er}$ in spring but only 31% in winter. However, for 2004-2015, TO and CC together
explained the greatest proportion of $H_{er}$ variability in winter and spring (67% and 63%) with
summer and autumn explaining 41% and 37% respectively.

The season-specific analysis of these data also showed that the size of the respective
contributions that TO and CC made to the variation in $H_{er}$ changed between the two periods. For
1991-2004, in spring and summer, TO explained 17% and 7% of the variability respectively
compared to the 65% and 63% contributions of CC. In winter and spring of 2004-2015, TO
explained 52% and 48% of the $H_{er}$ variability compared to a CC contribution of 15% for both
seasons.

When the effect of TO, CC and AOD were combined together for the period 2006-2015F, we
found a significant evidence that variation in $H_{er}$ could be partially explained by the changes in
TO and CC. However, there was a lack of clear evidence of any underlying dependence of AOD
on changes in $H_{er}$, although the effect of AOD on $H_{er}$ was positive. For the season-specific
analysis, when the AOD effect added to the model that contained the TO and CC effects, the
AOD effect was statistically significant only in winter (p=0.04), where the AOD effect on $H_{er}$ was
positive (0.18%) and the corresponding estimate for TO and CC was -1.44 and -1.04 respectively
over the period 2006-2015. The total variation explained in $H_{er}$ by these three factors together in
winter was 76% and of this total TO, CC and AOD contributed to the changes were 57%, 14%
and 5%.

This study demonstrated that for the first period 1991-2004, the CC effect on $H_{er}$ was greater
than that of TO, while for the second period 2004-2015 the opposite was found and the impact of
AOD was negligible, although the AOD effect was small and significant in winter. Our findings
from the first period partly agree with those from the Austrian study at Hoher Sonnblick over the
period 1997-2011 and the Spanish study on the Iberian Peninsula for the period 1985-2011.
Both studies reported that the significant increase in $H_{er}$ was attributed to changes in CC and
AOD and rather than TO (Román et al., 2015, Fitzka et al., 2012). The study over Canada,
Europe and Japan during 1995-2006 showed that the decline of AOD and significant increase in
TO, were associated with increased UV-B, although a non-significant trend with CC was found
(Zerefos et al., 2012). In contrast, the Belgian study reported that AOD had only a very small
impact on erythema UV dose whilst the impact from TO was strong (De Bock et al., 2014).
This study provides evidence that both the increasing trend in $H_{er}$ for 1991-2004 and the
decreasing trend in $H_{er}$ for 2004-2015 occur in response to variation in TO which exhibits a small
increasing tendency over these periods. CC plays an important role in the increasing trend in $H_{er}$
for 1991-2004 than TO. Whereas for 2004-2015, the decreasing trend in $H_{er}$ is less associated
with changes in CC and AOD.
**Acknowledgements**

We wish to thank Ruth Petrie and to the whole of the Centre for Environmental Data Analysis
(CEDA) for all their help and providing us the cloud cover data from the Benson station at
Oxfordshire. We also thank Iain H.Woodhouse, Judith Agnew and Judith Jeffrey and their staff
for establishing and maintaining the AERONET sites and also NERC Field Spectroscopy Facility
(FSF) used in this investigation. We would also like to express thanks to the Department for
Environment, Food and Rural Affairs (DEFRA) for making total ozone data available on their
website. The authors would like to thank three anonymous reviewers and co-editor for their
constructive comments.

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

**Table 1:** Daily $H_{er}$ (J m$^{-2}$) averaged over the whole period and for each season in Chilton during 1991-2015.

|                    | Min | Mean | Median | Stdev. | Max  |
|--------------------|-----|------|--------|--------|------|
| Whole data         | 10  | 1294 | 917    | 1179   | 5655 |
| Winter (Dec-Feb)   | 10  | 188  | 140    | 148    | 933  |
| Spring (Mar-May)   | 84  | 1606 | 1463   | 943    | 4880 |
| Summer (June-Aug)  | 212 | 2617 | 2552   | 944    | 5655 |
| Autumn (Sep-Nov)   | 23  | 746  | 540    | 618    | 2913 |

**Table 2**: Estimated trends (in %, $y^{-1}$) for $H_{er}$ with 95% confidence intervals (CI) at Chilton for two sub-periods: 1991-2004 and 2004-2015.

| | Study period | | | |
| --- | --- | --- | --- | --- |
| | 1991-2004 | | 2004-2015 | |
| | Estimated trend (95% CI) | p-value | Estimated trend (95% CI) | p-value |
| Monthly data | 1.01 (0.48; 1.54) | <0.001 | -1.35 (-1.98; -0.77) | <0.001 |
| Winter (Dec.-Feb.) | 1.29 (0.17; 2.41) | 0.03 | -1.08 (-3.14; 1.02) | 0.24 |
| Spring (Mar.-April) | 0.84 (-0.40; 2.05) | 0.22 | -0.88 (-2.10; -0.34) | 0.16 |
| Summer (June-Aug.) | 0.74 (-0.15; 1.76) | 0.09 | -1.67 (-2.48; -0.86) | <0.001 |
| Autumn (Sep.-Nov.) | 0.98 (-0.04; 2.00) | 0.07 | -1.56 (-2.68; -0.44) | 0.01 |

**Table 3**: Estimated trends (in %, y$^{-1}$) for TO with 95% CI at southern England for the monthly mean deviation data and for each season using various study periods.

| | Study period | | | | | |
| | 1991-2015 | | 1991-2004 | | 2004-2015 | |
| | Estimated trend (95% CI) | p-value | Estimated trend (95% CI) | p-value | Estimated trend (95% CI) | p-value |
|---|---|---|---|---|---|---|
| Whole data | 0.17 (0.09; 0.25) | <0.001 | 0.19 (-0.006; 0.38) | 0.06 | 0.28 (0.03; 0.53) | 0.03 |
| Winter | 0.43 (0.19; 0.67) | <0.001 | 0.31 (-0.20; 0.82) | 0.24 | 0.66 (-0.14; 1.59) | 0.10 |
| Spring | 0.15 (-0.02; 0.32) | 0.09 | 0.22 (-0.16; 0.64) | 0.29 | 0.06 (-0.41; 0.53) | 0.80 |
| Summer | 0.03 (-0.07; 0.13) | 0.52 | 0.02 (-0.21; 0.25) | 0.87 | 0.13 (-0.09; 0.35) | 0.26 |
| Autumn | 0.05 (-0.07; 0.23) | 0.27 | 0.05 (-0.30; 0.40) | 0.78 | 0.26 (-0.17; 0.69) | 0.24 |

**Table 4**: Estimated effect of TO ozone on $H_{er}$ with 95% confidence interval based on three study periods (CI).

| | 1991-2015 | | 1991-2004 | | 2004-2015 | |
|---|---|---|---|---|---|---|
| | Estimate (95% CI) | $r^2$ (%) | Estimate (95% CI) | $r^2$ (%) | Estimate (95% CI) | $r^2$ (%) |
| Whole data | -1.33 (-1.60; -1.06) | 25 | -1.18 (-1.57;-0.79) | 18 | -1.50 (-1.85; -1.15) | 33 |
| Winter | -0.94 (-1.37; -0.51) | 19 | -0.54 (-1.23; 0.15)# | 6 | -1.66 (-2.19; -1.13) | 52 |
| Spring | -1.88 (-2.39; -1.37) | 41 | -1.78 (-2.50; -1.06) | 37 | -1.87 (-2.54; -1.20) | 48 |
| Summer | -2.37 (-3.13; -1.61) | 34 | -2.47 (-3.35; -1.59) | 42 | -2.18 (-3.49; -0.87) | 24 |
| Autumn | -1.39 (-2.00; -0.78) | 21 | -1.57 (-2.37; -0.77) | 27 | -1.19 (-2.07; -0.31) | 17 |

#: $p = 0.12$;

**Table 5**: Estimated effect of CC on $H_{er}$ (%) with 95% CI, based on three study periods.

| | 1991-2015 | | 1991-2004 | | 2004-2015 | |
|---|---|---|---|---|---|---|
| | Estimate (95% CI) | $r^2$ (%) | Estimate (95% CI) | $r^2$ (%) | Estimate (95% CI) | $r^2$ (%) |
| Whole data | -0.95 (-1.09; -0.81) | 38 | -1.06 (-1.23;-0.89) | 48 | -0.82 (-1.04; -0.60) | 27 |
| Winter | -1.09 (-1.50; -0.68) | 27 | -0.96 (-1.47; -0.45) | 25 | -1.20 (-1.83; -0.57) | 29 |
| Spring | -1.05 (-1.30; -0.80) | 48 | -1.20 (-1.47; -0.93) | 66 | -0.99 (-1.38; -0.60) | 42 |
| Summer | -0.73 (-0.90; -0.54) | 46 | -0.92 (-1.14; -0.70) | 64 | -0.53 (-0.78; -0.28) | 31 |
| Autumn | -1.05 (-1.34; -0.76) | 41 | -1.15 (-1.48; -0.82) | 53 | -0.87 (-1.36; -0.38) | 26 |

**Table 6**: Estimated effect on $H_{er}$ with 95% CI from the combined effect of both TO and CC trend for the period 1991-2015.

|  | total ozone (95% CI) | cloud cover (95% CI) | $r^2$ (%) |
|---|---|---|---|
| Whole data | -1.03 (-1.25; -0.81) | -0.82 (-0.94; -0.70) | 51 |
| Winter | -0.85 (-1.22; -0.48) | -1.02 (-1.39; -0.65) | 42 |
| Spring | -1.41 (-1.81; -1.01) | -0.84 (-1.06; -0.62) | 68 |
| Summer | -1.38 (-2.09; -0.67) | -0.56 (-0.76; -0.36) | 55 |
| Autumn | -0.98 (-1.49; -0.47) | -0.92 (-1.19; -0.65) | 51 |

**Table 7**: Estimated effect on $H_{er}$ with 95% CI from the combined effect of TO and CC trend for two sub-periods: 1991-2004 and 2004-2015.

| | 1991-2004 | | | 2004-2015 | | |
|---|---|---|---|---|---|---|
| | cloud cover (95% CI) | total ozone (95% CI) | $r^2$ (%) | cloud cover (95% CI) | total ozone (95% CI) | $r^2$ (%) |
| Whole data | -0.97(-1.13; -0.81) | -0.79 (-1.26; -0.68) | 55 | -0.65 (-0.85; -0.45) | -1.25 (-1.56; -0.94) | 49 |
| Winter | -0.96 (-1.45; -0.47) | -0.55 (-1.14; 0.04)* | 31 | -0.89 (-1.34; -0.44) | -1.45 (-1.92; -0.98) | 67 |
| Spring | -1.04 (-1.26; -0.82) | -1.26 (-1.67; -0.85) | 82 | -0.66 (-1.01; -0.31) | -1.36 (-1.99; -0.73) | 63 |
| Summer | -0.73 (-0.97; -0.49) | -1.17 (-1.56; -0.78) | 70 | -0.41 (-0.76; -0.06) | -1.45 (-2.70; -0.20) | 41 |
| Autumn | -1.00 (-1.31; -0.69) | -1.03 (-1.64; -0.42) | 64 | -0.77 (-1.24; -0.30) | -0.97 (-1.75; -0.19) | 37 |

*: p-value=0.07

Figure 1:

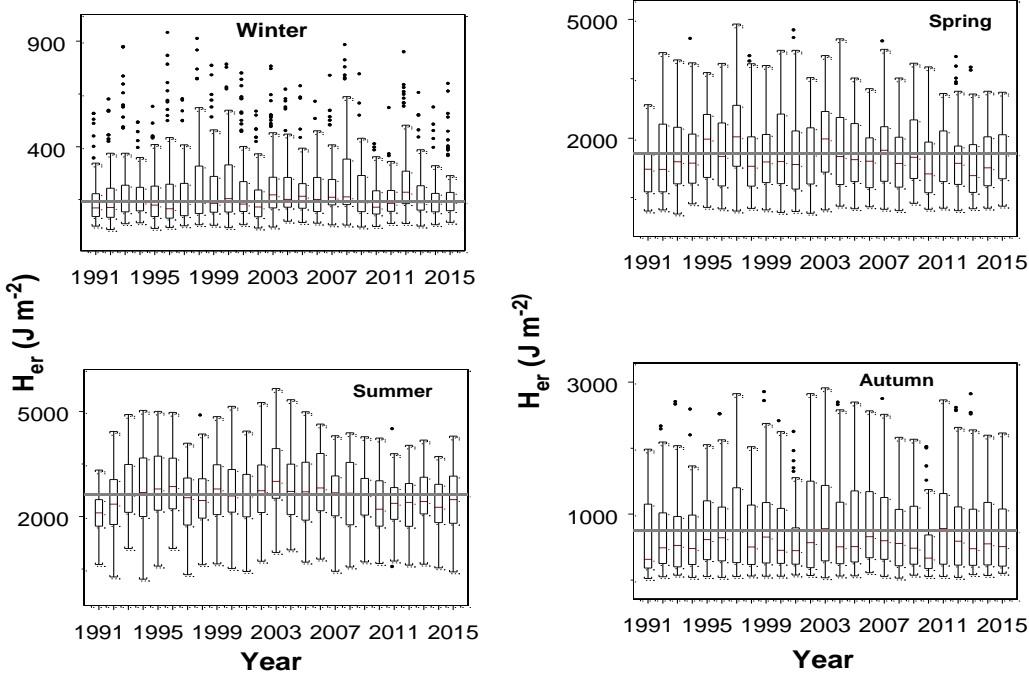

Figure 2:

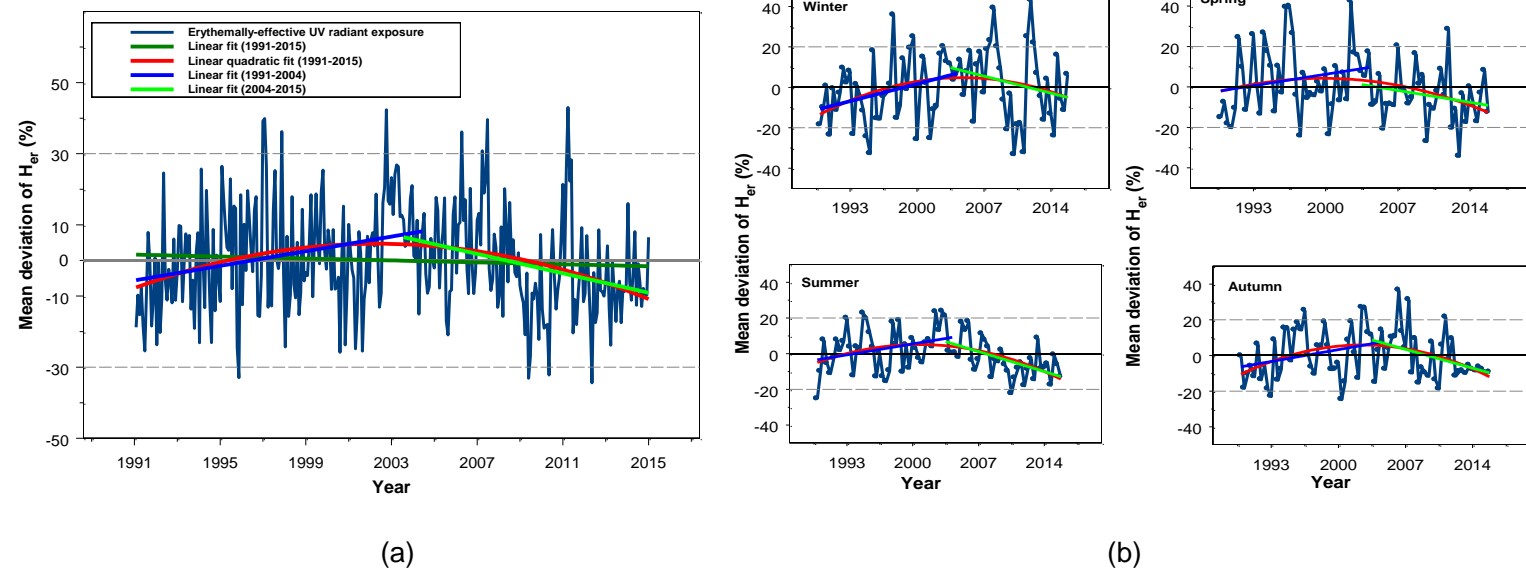

(a)                                                           (b)

Figure 3:

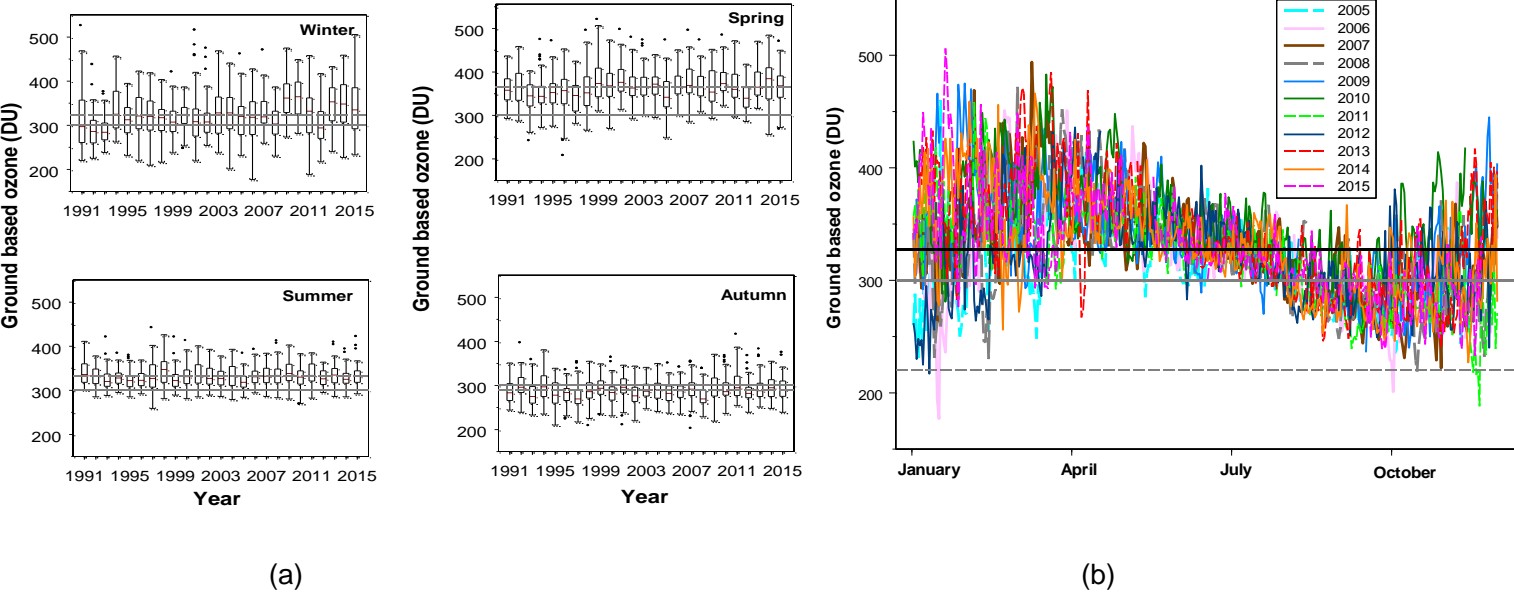

(a)                                                                   (b)

Figure 4:

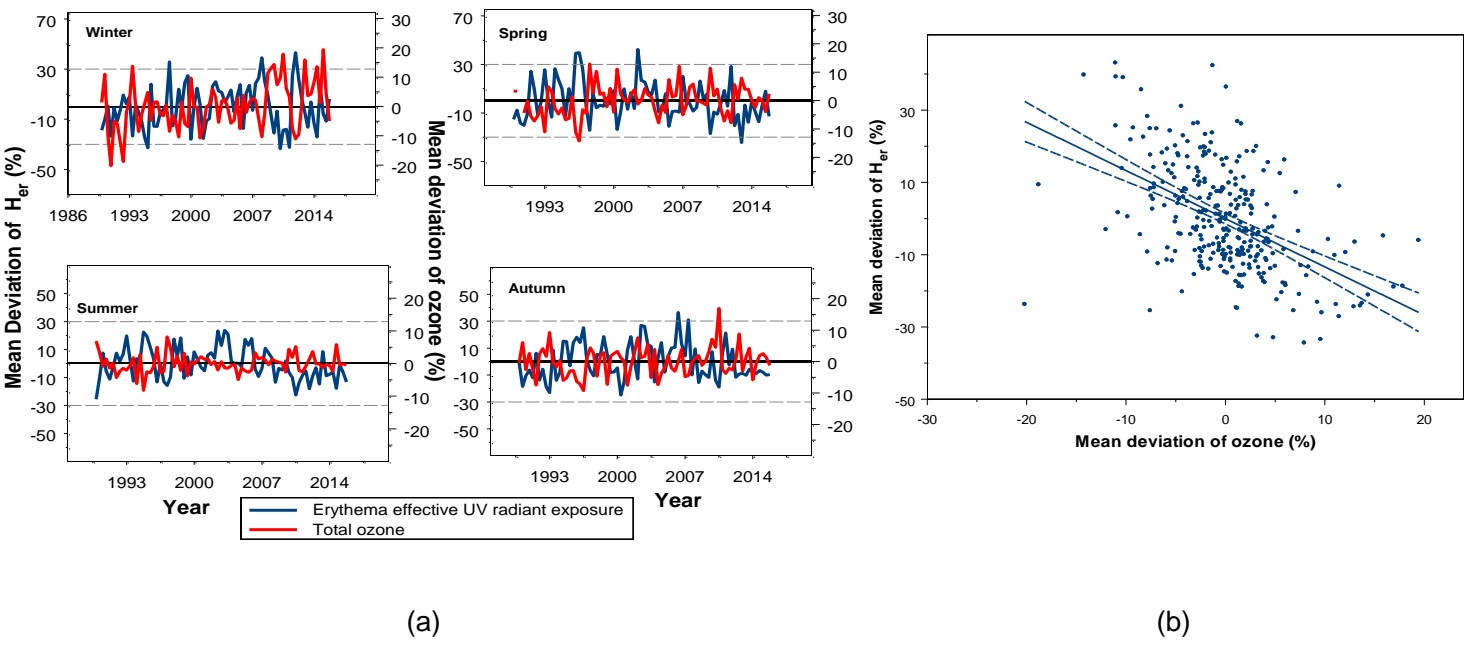

(a)                                                                    (b)

Figure 5:

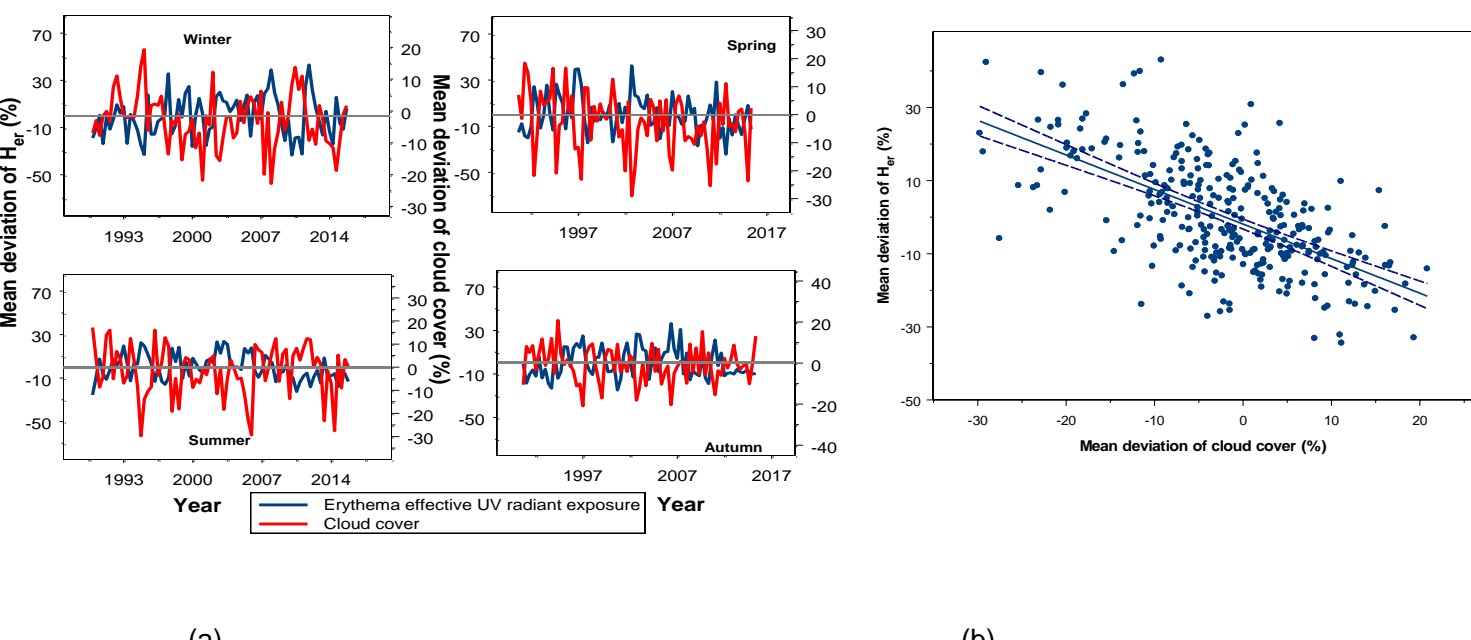

(a)                                                                          (b)