# Peer review of "Relationship between erythema effective UV radiant exposure, total ozone, cloud"

_Atmospheric Chemistry and Physics, 2018_

## Referee Comment (RC1) · Anonymous Referee #3 · 20 Sep 2018

The manuscript by Hunter et al is clearly written and well organized. The authors study the short- and long-term changes of the daily erythemal doses over Chilton relative to the changes of total ozone and cloudiness, using a very long record of ground-based UV measurements. The study is a good contribution for the UV community. However, there are some issues that have to be addressed prior to the publication of the study.

Page 2, lines 58 – 63: Quantification of the effect of each of these factors is not easy, because of the complex interaction between them and the solar UV radiation. For example, the effect of clouds changes depending on the presence of aerosols (and is different for different types of aerosols). At least a discussion pointing out these

complex interactions should be added here.

The authors treat the effects of changes in ozone and cloudiness on erythemal irradiance as linear and independent to each other. However, they are nor linear, neither completely independent to each other. I suggest that a short discussion explaining why the particular methodology was chosen and what are the limitations/uncertainties due to its use should also be added in the introduction.

Section 2.3 (Estimating trends): The authors have not taken into account the variations of QBO and solar cycle in the analysis. Both phenomena are periodical and affect the variability of total ozone and UV-B radiation. Since these phenomena affect the results of the study, their effect should be either removed or at least quantified. Another, useful information which should be added here is the treatment of gaps in the series i.e.: - Is there a minimum number of available days below which a month is not taken into account in the analysis? - What if some measurements are missing during a day? Is there any particular criterion used in order to include a particular day in the analysis?

Section 3: (Figures 1 and 3): How were the measurements outside the whiskers classified as outliers (i.e. which criteria were used in order to characterize a measurement outlier)? P5, L183: what does the word "corrected" means? How and for what was the monthly deviation corrected?

Section 4: The results presented in this manuscript are also in good agreement with the results of Fountoulakis et al (2016) ("Short- and long-term variability of spectral solar UV irradiance at Thessaloniki, Greece: effects of changes in aerosols, total ozone and clouds") where a turning point in the trends of UV irradiance is reported on 2006. Can the authors comment the similar behavior of UV radiation at the two sites (between which the distance is very long, and the climatological conditions differ importantly)?
* * *

---

## Referee Comment (RC2) · Anonymous Referee #1 · 4 Oct 2018

General comments:

This paper provides a sound study on long-term trends over 25 years of solar ultraviolet radiation measurements and the observations are related to ozone and cloud cover. The scientific content of the publication is certainly worth to be published in ACP.

The submitted paper is well written and organized and the data are fully described.

However, the statistical methods are based on tests, which might not be well known to the reader. I suggest including a small summary of the statistical tests or at least to citing the publications, were the methods are fully described.

[Figure]

The paper can be published with this minor revision requested.

Specific comments:

Section 2.3:

Are the tests performed with all available data points? What would be the impact on the results if only a randomly chosen subset of data are selected for the tests? In other words: What is the variation of the statistical results, when a smaller of data-points are used for the calculation. Or: how robust are the statistical results.

Smaller issues:

The abstract basically describes the intention and results of the study – no changes.

The text is well written and no major typos have been detected.

---

## Referee Comment (RC3) · Anonymous Referee #2 · 4 Oct 2018

This manuscript explores the changes in erythema effective UV radiant exposure over a 25 year period, and the associated changes in total ozone and cloud cover that might be expected to influence UV radiation at the ground. This is a significant time series for ground-based UV radiation measurements and as such the results are instructive. The ozone and cloud cover data have been taken from longer datasets for stations relatively close to Chilton, the location for the UV measurements. The work is well presented but appears as a statistical exercise somewhat lacking in atmospheric interpretation. It raises a number of queries that must be addressed before publication of a final paper.

Section 2.1

The previous paragraph states that monthly UV doses are considered in the manuscript. Section 2.1 then details how a daily dose was calculated. Please specify how a monthly dose was then determined – is it the sum of all days in the month, or the average of all days in the month (that is it becomes a mean daily dose for the month). How was missing data treated? Was there a limit to the number of missing hours allowed for calculation of a daily dose, and similarly what were restrictions on missing days in determining a monthly dose? The same questions apply to the external datasets that have been used for ozone and cloud cover. What were the minimum number of years that contributed to the overall monthly average for each of the 3 data sets?

Please provide a brief statement on the traceability and stability of calibration of the radiometers over the 25 year period. What is the associated uncertainty in the measurements and how can you be sure that there has been no drift, short- or long-term, in the measurement system?

Section 2.3 Seasonal variations have been removed from the data, but have longer term cycles been considered e.g. QBO and 11 (or 22) year solar cycle?

Please explain, or at least reference, the statistical techniques used (DW, MK, SS).

Section 3.1 Figure 1 – how were 'outliers' identified? In all seasons except winter the outliers from one year are clearly within the bounds of acceptable data for other years, so why have these data points been excluded? If they were beyond possibility for the site then there would be good reason to exclude the points, but this is not the case. In winter there are a large number of outliers – how did you determine that these data were unreliable? Please provide a clear justification for removing what appear to be valid data points from the analysis.

Define seasons i.e. which months have been used as 'winter'

Section 3.2 The annual ozone cycle is as one would expect at these latitudes. Comment on this and causes of e.g. low ozone events / particular occurrences e.g. in 2011. Note summer ozone (when UV is high) has very small and non-significant trends over any time period. The significant ozone trends in winter will influence the very low UV doses at that time of year, but have little practical influence on overall annual dose of UV. This fact is somewhat lost in dealing only in percentage deviations from average, where the winter % has the same weight as the summer %. Further comments on the implications for absolute UV doses are needed throughout.

Figure 3 – again please justify 'outliers'.

Fig 3b – what are the black line and the grey dashed line? The latter is not the mean value, as described in the text.

Section 3.3 Line 300 – comment on this with respect to Radiation Amplification Factors. Also comment on why RAF apparently changes with season or with period considered.

Section 3.5 Line 366-7 – qualify this statement, it is not necessarily a global truth. Also further down the paragraph you show that for a 1% change in cloud or ozone the response in H is greater for ozone.

Section 4 Lines 430 – 444 This does not produce a convincing argument for the analysis in this manuscript vs that of the previous publication. Both are described as 'best/better described by two linear trends'. Since both works use the same data set, how can the two linear trend selections be so different in the pivot point used to change from one trend to the next? This needs further justification. The overall change (full data set) should be the same for both analyses since the underlying data is the same. Is this the case?

Section 4.4 – discussion on aerosols. This is rather inconclusive. If AOD has been stable at Chilton then changes in aerosol/pollution cannot explain any changes in H. What is left as an explanation?

Lines 642-8 This (and the similar paragraph in the abstract) is almost counter-intuitive

in trying to manufacture associations between small changes in H, ozone and cloud cover. 1991-2004 has increased H associated with decreased cloud and no significant change in ozone (section 4, the abstract says there is an upward trend in ozone). 2004 – 2015: section 4 says there is a slowdown in the upward trend in H, and in the next sentence says there is a significant decrease in H. Both cannot be correct. The abstract only mentions a decrease in H. This is associated with a marginal upward trend in ozone and no significant change in cloud.

The abstract and discussion should be made consistent with each other. The abstract implies that both increasing and decreasing H occur at the same time as increasing ozone, but increasing H is more strongly linked to reductions in cloud cover, while there is no significant change in cloud over the period that H is reducing. Added to which all changes are small and occur within a very variable signal. Such a comment in the abstract, that all changes are small and some are not statistically significant, seems necessary.

---

## Author Comment (AC1) · 26 Oct 2018

Response to referee comments:
The manuscript by Hunter et al is clearly written and well organized. The authors study the short- and long-term changes of the daily erythemal doses over Chilton relative to

the changes of total ozone and cloudiness, using a very long record of ground-based UV measurements. The study is a good contribution for the UV community. However, there are some issues that have to be addressed prior to the publication of the study.

Page 2, lines 58 – 63: Quantification of the effect of each of these factors is not easy, because of the complex interaction between them and the solar UV radiation. For example, the effect of clouds changes depending on the presence of aerosols (and is different for different types of aerosols). At least a discussion pointing out these complex interactions should be added here. Response: Done; see Line 68-69. The authors treat the effects of changes in ozone and cloudiness on erythemal irradiance as linear and independent to each other. However, they are nor linear, neither completely independent to each other. I suggest that a short discussion explaining why the particular methodology was chosen and what are the limitations/uncertainties due to its use should also be added in the introduction.

Response: Statistical linear or non-linear models have been used in a number of applications for the ground UV radiation research (Zerefos et al. 2012; V De Bock et. Al 2014; Smedley et al. 2012). These models statistically relate ground-based measurements of surface UV irradiance as dependent variables and ozone and cloud cover as independent variables. Thus, these models were also used here to make comparison with the published results in the literature. Certainly some of the methods are comparable, e.g. linear versus non-linear models, but as shown here overall estimate and findings were similar using both models. Thus, we do not think discussion regarding statistical modelling issues needed in the introduction section.

Section 2.3 (Estimating trends): The authors have not taken into account the variations of QBO and solar cycle in the analysis. Both phenomena are periodical and affect the variability of total ozone and UV-B radiation. Since these phenomena affect the results of the study, their effect should be either removed or at least quantified.

Response: We have not taken into account the QBO (quasi-biennial oscillation) or the

solar cycle in the analysis. The explanation for this is as follows: Response: We agree with the review that the quasi-biennial oscillation (QBO) and the 11-year solar cycle are also factors that affect UV levels, particularly through their impact on ozone and clouds (Den Outer, 2005). Since the period of the QBO is approximately 2.3 years it affects short term variability rather than long term trends (Harris et al., 2008, Den Outer et al., 2005). This fluctuation is small in comparison to the 25 year timescale being analysed in this paper. Relevant text has been added into section 2.3 see Line 157-163. The 11-year solar cycle has a longer period and therefore has the potential to impact long term trends, however its effect on erythema effective UV levels is small (Den Outer, 2005, Diffey, 2002). We have investigated whether solar activities (the 11-year solar cycle is a cycle of sunspot activity, i.e. the number of sunspots) affect the changes in UV radiation at Chilton. The relationship between UVR values and total sunspot numbers was studied and the relationship between sunspots and UVR also appeared to be reciprocal; UVR being high when sunspot is low, and vice versa. However, using t-test, the correlation between UVR values at Chilton and sunspot numbers was not statistically significant (P=0.27) during the time period from 1991 to 2015. Thus, we have decided not to include the results in this manuscript.

References: Den Outer, P.N., Slaper, H. and Tax, R.B., 2005. UV radiation in the Netherlands: Assessing long‐term variability and trends in relation to ozone and clouds. Journal of Geophysical Research: Atmospheres, 110(D2). Harris, N.R., Kyrö, E., Staehelin, J., Brunner, D., Andersen, S.B., Godin-Beekmann, S., Dhomse, S., Hadjinicolaou, P., Hansen, G., Isaksen, I. and Jrrar, A., 2008. Ozone trends at northern mid-and high latitudes–a European perspective. In Annales Geophysicae (Vol. 26, No. 5, pp. 1207-1220). Diffey, B.L., 2002. Sources and measurement of ultraviolet radiation. Methods, 28(1), pp.4-13.

Another, useful information which should be added here is the treatment of gaps in the series i.e.: -Is there a minimum number of available days below which a month is not taken into account in the analysis? - What if some measurements are missing during

a day? Is there any particular criterion used in order to include a particular day in the analysis?

Response: There is no minimum number of available days in a month. The criterion for including a day in the analysis is that the data are complete during the relevant period of time for each day – that is, that no data points are missing from 30 minutes before sunrise to 30 minutes after sunset (Fig1).

Response: If any data points are missing during the relevant period of time for each day (30 minutes before sunrise to 30 minutes after sunset) the whole day is rejected. 95.7% of months have 5 missing days or less. 85.3% of months have 1 missing day or less and 68% of months have no missing days at all. Thus, we did not think that it was necessary to include in this manuscript.

Section 3: (Figures 1 and 3): How were the measurements outside the whiskers classified as outliers (i.e. which criteria were used in order to characterize a measurement outlier)? P5, L183: what does the word "corrected" means? How and for what was the monthly deviation corrected?

Response: In statistical term, outliers known as the extreme data points are outside the typical pattern of the other data sets. It is possible to delete outliers from the data set before analysis or use non-parametric statistical methods that are less influenced by outliers. We did not remove them because these points could be real measurements. The UV dose values might have fluctuated more especially in winter at this site due to natural variations which affect UV dose, in particular extremely low total ozone often occurs in winter. We should also bear in mind that the winter data had the lowest UV dose level among the rest of seasons in Chilton. The text has been revised in section 3.1 (Line 208-212). The use of word "corrected" is confusing and it has been removed throughout the paper.

Section 4: The results presented in this manuscript are also in good agreement with the results of Fountoulakis et al (2016) ("Short- and long-term variability of spectral solar

UV irradiance at Thessaloniki, Greece: effects of changes in aerosols, total ozone and clouds") where a turning point in the trends of UV irradiance is reported on 2006. Can the authors comment the similar behavior of UV radiation at the two sites (between which the distance is very long, and the climatological conditions differ importantly)? Response: Done; see Line 527-533.

[Figure]

[Figure]

**Fig. 1.**

---

## Author Comment (AC2) · 26 Oct 2018

Response to referee comments:
General comments: This paper provides a sound study on long-term trends over 25 years of solar ultraviolet radiation measurements and the observations are related to

ozone and cloud cover. The scientific content of the publication is certainly worth to be published in ACP.

The submitted paper is well written and organized and the data are fully described.

However, the statistical methods are based on tests, which might not be well known to the reader. I suggest including a small summary of the statistical tests or at least to citing the publications, were the methods are fully described.

Response: We have now extended the paragraph and some references have been added in section 2.3 (see Line 173-184).

The paper can be published with this minor revision requested.

Specific comments: Section 2.3: Are the tests performed with all available data points? What would be the impact on the results if only a randomly chosen subset of data are selected for the tests? In other words: What is the variation of the statistical results, when a smaller of data-points are used for the calculation. Or: how robust are the statistical results.

Response: We use all available data covering period 1991-2015 for UV dose, total ozone and cloud cover data. The aim of the study is to investigate the long term variability, studying a small subset of the data can give misleading results due to lack of statistical power. Conclusions from short-term changes cannot be extrapolated to long term variations because UV dose can undergo rapid fluctuations at any location due to ozone, weather, cloud cover etc. affecting the UV dose occurring at any time.

Smaller issues: The abstract basically describes the intention and results of the study – no changes. The text is well written and no major typos have been detected.

---

## Author Comment (AC3) · 26 Oct 2018

Response to referee comments:
This manuscript explores the changes in erythema effective UV radiant exposure over a 25 year period, and the associated changes in total ozone and cloud cover that might

be expected to influence UV radiation at the ground. This is a significant time series for ground-based UV radiation measurements and as such the results are instructive. The ozone and cloud cover data have been taken from longer datasets for stations relatively close to Chilton, the location for the UV measurements. The work is well presented but appears as a statistical exercise somewhat lacking in atmospheric interpretation. It raises a number of queries that must be addressed before publication of a final paper.

Section 2.1 The previous paragraph states that monthly UV doses are considered in the manuscript. Section 2.1 then details how a daily dose was calculated. Please specify how a monthly dose was then determined – is it the sum of all days in the month, or the average of all days in the month (that is it becomes a mean daily dose for the month). How was missing data treated? Was there a limit to the number of missing hours allowed for calculation of a daily dose, and similarly what were restrictions on missing days in determining a monthly dose? The same questions apply to the external datasets that have been used for ozone and cloud cover. What were the minimum number of years that contributed to the overall monthly average for each of the 3 data sets?

Response: Monthly mean data for UV doses, total ozone and cloud cover are averaged from summing all daily values by month and then dividing by the number of days in the month. There are missing measurements in the daily data sets, but they are not significant to report in the manuscript; missing values are accounted only about 3% of the daily recorded UV dose and about 8% of the daily recorded total ozone and there was no missing values for cloud cover data. The statistical package used here excludes all missing values for the analysis.

Please provide a brief statement on the traceability and stability of calibration of the radiometers over the 25 year period. What is the associated uncertainty in the measurements and how can you be sure that there has been no drift, short- or long-term, in the measurement system?
Response: The broadband detectors measuring erythema effective UV radiation are calibrated annually using a double-grating spectroradiometer. The spectroradiometer is calibrated and is traceable to national standards. The daily radiant exposure for 22 clear days during May–October between 2003 and 2015 was compared to the daily radiant exposure from the double-grating spectroradiometer and the data from the broadband detectors was found to be within 10% of the spectroradiometer data on all these days. (Hooke, 2017) Relevant text has been added into section 2.1 (see Line 100-105).

Reference: Hooke, R.J., Higlett, M.P., Hunter, N. and O'Hagan, J.B., 2017. Long term variations in erythema effective solar UV at Chilton, UK, from 1991 to 2015. Photochemical & Photobiological Sciences, 16(11), pp.1596-1603.

Section 2.3 Seasonal variations have been removed from the data, but have longer term cycles been considered e.g. QBO and 11 (or 22) year solar cycle?

Response: We have not taken into account the QBO (quasi-biennial oscillation) or the solar cycle in the analysis. The explanation for this is as follows: We agree with the review that the quasi-biennial oscillation (QBO) and the 11-year solar cycle are also factors that affect UV levels, particularly through their impact on ozone and clouds (Den Outer, 2005). Since the period of the QBO is approximately 2.3 years it affects short term variability rather than long term trends (Harris et al., 2008, Den Outer et al., 2005). This fluctuation is small in comparison to the 25 year timescale being analysed in this paper. Relevant text has been added into section 2.3 (see Line 157-163).

The 11-year solar cycle has a longer period and therefore has the potential to impact long term trends, however its effect on erythema effective UV levels is small (Den Outer, 2005, Diffey, 2002). We have investigated whether solar activities (the 11-year solar cycle is a cycle of sunspot activity, i.e. the number of sunspots) affect the changes in UV radiation at Chilton. The relationship between UVR values and total sunspot numbers was studied and the relationship between sunspots and UVR also appeared

to be reciprocal; UVR being high when sunspot is low, and vice versa. However, using t-test, the correlation between UVR values at Chilton and sunspot numbers was not statistically significant (P=0.27) during the time period from 1991 to 2015. Thus, we have decided not to include the results in this manuscript.

References: Den Outer, P.N., Slaper, H. and Tax, R.B., 2005. UV radiation in the Netherlands: Assessing long‐term variability and trends in relation to ozone and clouds. Journal of Geophysical Research: Atmospheres, 110(D2). Harris, N.R., Kyrö, E., Staehelin, J., Brunner, D., Andersen, S.B., Godin-Beekmann, S., Dhomse, S., Hadjinicolaou, P., Hansen, G., Isaksen, I. and Jrrar, A., 2008. Ozone trends at northern mid-and high latitudes–a European perspective. In Annales Geophysicae (Vol. 26, No. 5, pp. 1207-1220). Diffey, B.L., 2002. Sources and measurement of ultraviolet radiation. Methods, 28(1), pp.4-13.

Please explain, or at least reference, the statistical techniques used (DW, MK, SS).

Response: We have now extended the paragraph and some references have been added in section 2.3 (see Line 173-184).

Section 3.1 Figure 1 – how were 'outliers' identified? In all seasons except winter the outliers from one year are clearly within the bounds of acceptable data for other years, so why have these data points been excluded? If they were beyond possibility for the site then there would be good reason to exclude the points, but this is not the case. In winter there are a large number of outliers – how did you determine that these data were unreliable? Please provide a clear justification for removing what appear to be valid data points from the analysis. Response: In statistical term, outliers known as the extreme data points are outside the typical pattern of the other data sets. It is possible to delete outliers from the data set before analysis or use non-parametric statistical methods that are less influenced by outliers. We did not remove them because these points could be real measurements. The UV dose values might have fluctuated more especially in winter at this site due to natural variations which

affect UV dose, in particular extremely low total ozone often occurs in winter. We should also bear in mind that the winter data had the lowest UV dose level among the rest of seasons in Chilton. The text has been revised in section 3.1 (Line 208-212).

Define seasons i.e. which months have been used as 'winter'

Response: Done, see Table 1 (winter: December, January and February).

Section 3.2 The annual ozone cycle is as one would expect at these latitudes. Comment on this and causes of e.g. low ozone events / particular occurrences e.g. in 2011. Note summer ozone (when UV is high) has very small and non-significant trends over any time period. The significant ozone trends in winter will influence the very low UV doses at that time of year, but have little practical influence on overall annual dose of UV. This fact is somewhat lost in dealing only in percentage deviations from average, where the winter % has the same weight as the summer %. Further comments on the implications for absolute UV doses are needed throughout.

Response: Text has been added in section 3.2 see Line 317-321 and also section 3.3, see Line 337-338.

Figure 3 – again please justify 'outliers'.

Response: Done, see section 3.2, Line 281-282

Fig 3b – what are the black line and the grey dashed line? The latter is not the mean value, as described in the text.

Response: Done, see section 3.2, Line 285-287

Section 3.3 Line 300 – comment on this with respect to Radiation Amplification Factors. Also comment on why RAF apparently changes with season or with period considered.

Response: Text has been added in various sections regarding the Radiation Amplification Factor (RAF). It is in section 3.3 (Line 347-348, 358-360 and 368-371), section 3.4 (Line 429-430 and 448-450) and section 4.4 (Line 662-667) and in Abstract.

Section 3.5 Line 366-7 – qualify this statement, it is not necessarily a global truth. Also further down the paragraph you show that for a 1% change in cloud or ozone the response in H is greater for ozone. Response: We have now deleted this sentence in Line 366-7 in section 3.5.

Section 4 Lines 430 – 444 This does not produce a convincing argument for the analysis in this manuscript vs that of the previous publication. Both are described as 'best/better described by two linear trends'. Since both works use the same data set, how can the two linear trend selections be so different in the pivot point used to change from one trend to the next? This needs further justification. The overall change (full data set) should be the same for both analyses since the underlying data is the same. Is this the case?

Response: Yes, the data used here are the same data set that previously published. In our previous published study, the analyses were based on annual mean anomaly data from the daily data, while the analyses performed here are monthly mean deviation data from the monthly data. Although annual data and monthly means show similar pattern, we have decided using monthly data in order to examine the effects of total ozone and cloudiness changes on the Her. Some relevant text has been added in section 2.3, see Line 151-155.

With regard to the pivot point, the previous paper split the time series due to geophysical phenomena – that is, the ozone turning point in the mid-1990s (WMO, 2014). This paper splits the time series according to statistical analysis. Clarification has been made, see section 4.1, Line 490-498.

Ref: WMO, Scientific Assessment of Ozone Depletion: 2014, World Meteorological Organisation (WMO), Geneva, Switzerland, 2014.

Section 4.4 – discussion on aerosols. This is rather inconclusive. If AOD has been stable at Chilton then changes in aerosol/pollution cannot explain any changes in H. What is left as an explanation?

Response: We do not have aerosol data to clarify your comments. It may be attributed to the variability in weather condition from climate change.

Lines 642-8 This (and the similar paragraph in the abstract) is almost counter-intuitive in trying to manufacture associations between small changes in H, ozone and cloud cover. 1991-2004 has increased H associated with decreased cloud and no significant change in ozone (section 4, the abstract says there is an upward trend in ozone). 2004 – 2015: section 4 says there is a slowdown in the upward trend in H, and in the next sentence says there is a significant decrease in H. Both cannot be correct. The abstract only mentions a decrease in H. This is associated with a marginal upward trend in ozone and no significant change in cloud. The abstract and discussion should be made consistent with each other. The abstract implies that both increasing and decreasing H occur at the same time as increasing ozone, but increasing H is more strongly linked to reductions in cloud cover, while there is no significant change in cloud over the period that H is reducing. Added to which all changes are small and occur within a very variable signal. Such a comment in the abstract, that all changes are small and some are not statistically significant, seems necessary.

Response: Done, see Abstract Line 28-33 and section 4.4, Line 718-723.

---

## Author Comment (AC4) · 26 Oct 2018

[revised manuscript text omitted]

(a)

(b)

Figure 3:

[Figure]

(a)                                                                                             (b)

Figure 4:

[Figure]

(a)                                                                                              (b)

Figure 5:

[Figure]

(a)                                                          (b)

---

## Author Comment (AC6) · 26 Oct 2018

The comment was uploaded in the form of a supplement:
https://www.atmos-chem-phys-discuss.net/acp-2018-828/acp-2018-828-AC6-supplement.pdf

---

## Author Comment (AC8) · 30 Oct 2018

The comment was uploaded in the form of a supplement:
https://www.atmos-chem-phys-discuss.net/acp-2018-828/acp-2018-828-AC8-supplement.pdf

---

## Author Comment (AC9) · 30 Oct 2018

The comment was uploaded in the form of a supplement:
https://www.atmos-chem-phys-discuss.net/acp-2018-828/acp-2018-828-AC9-supplement.pdf

---

## Editor Decision (ED1)

Based on the answer to the reviewers and the new document, I still have some major issues with major aspects of the used methodology and the interpretation of the results. I have got additional comments from one of the reviewers based on the new text and I have combined them with my comments that can be found below.

In long term series studies three major aspects have to be clear in order to ensure valuable results

- The data quality and the related uncertainty based on the quality assurance, quality control and calibration procedures that have been followed
- The methodology of treating/averaging/checking the datasets
- The interpretation of the results and the derived changes (here in UV) based on presented changes of the factors affecting UV.

More specific:

Abstract

I think that the included sentences such as: "All these changes are small and occur within a very variable signal." Plus the fact that is difficult to determine e.g. 1% changes with instrumentation that is just "better than 10%" have to be included here.

The abstract could be much more clear I suggest deleting the whole section starting with the new RAF text on line 19 up to line 27 and just continue with the last paragraph of new text (plus the comment above).

83 UV data

What was answered in one reviewer about not having gaps in the UV data series have to be included when describing the UV data section. It is amazing 25 years of UV data without gaps. Is the spectroradiometer calibration performed on site ?

Reading the argument from lines 100-105 it shows that in the second half of the 25 year period the max drift there could have been is 20% (+10% to -10%). The uncertainty in the data for all weather conditions and across the complete time period should be provided.

Also, after annual calibration, was data corrected if a change in the instrument was seen, and if so how? By linear interpolation back to the previous calibration, or some other method, or was the calibration only corrected going forward?

107 Ozone data

Ozone data could be continuous but Dobson and Brewer monitoring schedule/data availability depends on solar zenith angle and mostly cloud conditions. So some details on these data will help the reader to understand the use of the time series.

141 cloud averaging

A use of a constant 3 hour/day averaging of cloud coverage for assessing their impact on daily solar UV changes includes the uncertainty related with the cloud changes in the remaining daytime. This should be mentioned. A more realistic solution would be to weight the cloud amount for every hour with the percentage of cloudless sky UV for the specific hour versus daily cloudless UV. However, as this requires a lot of additional work the introduced uncertainty can just be mentioned.

144 Statistics

There is something that I can not understand.. statistically. How is it possible in a time series with no gaps the Hooke results that are based on summing up all days for a year and then calculate yearly anomalies and trends to be different than averaging daily values for each month and then calculating monthly anomalies and then trends. My impression is that the sum of the monthly anomalies in a year has to match the yearly anomaly calculated by Hook for the same year (with very small differences due to the small differences of the number of days in a month).

Outliers

More information on objective algorithms on rejecting outliers from the analysis should be provided.

The new text (lines 213-216) is contrary to the objective of the paper. If looking for underlying reasons for a trend (ozone, cloud) then data that may be particularly high/low due to ozone or cloud should not be removed from the dataset. They can be removed it if instrument problems have been identified. Otherwise the outliers seem arbitrary, and how can data be an outlier one year when it is well within the whisker value for another year?

Ozone and UV seasonal trends

For the second period where clouds are almost constant and ozone plays the only role in the UV trends: the same amount of ozone changes has different effect in the UV for different seasons due to the differences in the related air masses. More specific constant ozone trends during all seasons would theoretically lead to higher UV trends in winter than in the

summer months (in percent). From tables 2 and 3 seems that the opposite was found. Is there any explanation for this ?

Aerosols

The atmospheric related reasons for UV changes can be clouds, ozone, aerosol optical depth, other aerosol optical parameters, albedo, other traces gases, … (more or less ranked here based on their importance) changes. So if there are no data other than clouds and ozone you have to clarify it.

For the case of aerosols Zerefos et al., 2012 as you mention, presented changes also in non urban areas. I think that Aeronet/Chilbolton data and satellite data can be used in order to provide a hint on current speculations about a negligible 25 year effect of aerosols on this time series.

I could fast download 1 by 1 degree data around Chilton form MODIS Aqua shown below.

In addition Aeronet level 1.5 from Chilbolton station.

Modis/Aqua shows an AOD decreasing trend in the order of ~-0.003 or ~-1.5% per year. And Aeronet ~-0.007 or ~3% change per year.

This figure is a bit rough in terms of spatial resolution for Modis/Aqua and the data have been just plotted as they are with no checks at all. But some work on this aspect (e.g. lower spatial resolution Modis data or just use of the Aeronet data) can provide some more insights on the aerosol issue for the second period. Aeronet 1.5 level data also I just plotted the NASA site existing monthly mean data. They represent Chilbolton station/area.

[Figure]

If you end up on similar results then more or less results for this period agree with Zerefos et al., 2012 that state that there is a turning point that (for constant cloudiness) that the ozone increase masks the aerosol (slower rate than before) decrease for mid-latitudes.

figures

The quality of the figure 3b in the paper is not good it needs some improvement on the submitted figure format.

I would suggest to move some of the sections describing previous works related with UV vs clouds, ozone, aerosols in the introduction section and summarize the agreement/disagreement etc findings of your work compared with the mentioned publications in the last section.

One of the native English speakers on the author list should go through the new version of the manuscript and correct the grammar – particularly, but not limited to, the new text.

---

## Author Response (AR2)

**Response to Co-Editor comments:**

**Comment on "Relationship between erythema effective UV radiant exposure, total ozone and cloud cover in southern England UK: 1991–2015" by Nezahat Hunter et al.**

We welcome the Co-Editor's comments. Here are our replies to the issues raised. The manuscript has been changed throughout.

Please note that one of my colleagues surname has changed to "Rebecca. J. Rendell".

Based on the answer to the reviewers and the new document, I still have some major issues with major aspects of the used methodology and the interpretation of the results. I have got additional comments from one of the reviewers based on the new text and I have combined them with my comments that can be found below.

In long term series studies three major aspects have to be clear in order to ensure valuable Results

- The data quality and the related uncertainty based on the quality assurance, quality control and calibration procedures that have been followed

The data quality of the UV radiant exposure data is clearly beyond the scope of this paper. Regarding data quality and uncertainties, these have already been discussed in the following publications by Hooke et al. (2018) and Hooke et al. (2017). Further brief explanations have been also made in the text to clarify specific issues.

Detailed description about detectors and total ozone measurements in the Camborne and Reading sites have already been introduced in the published study by Smedley et al (2012). We have also investigated satellite-based total ozone data from the OMI data products in Reading and also in Chilton for the period (2005-2015). Both ground and satellite-based datasets were compared and the results revealed that the ground-based data in Reading are almost identical to those values from the satellite-based measurements in Reading and Chilton, but these results are not shown in the manuscript.

Cloud cover data have been collected together with other meteorological variables by the Met Office Hadley Centre in the UK and a detail description of the data has already been published elsewhere (Dunn et al. 2012 & 2014). The data were obtained from the Centre for Environmental Data Analysis (CEDA). The detailed information about data quality, uncertainties and calibration procedure in cloud cover measurements, again are outside the scope of this paper.

- The methodology of treating/averaging/checking the datasets

As a trained statistician with many years' experience, I believe the methodology and data analyses carried out here have followed the correct statistical procedures.

- The interpretation of the results and the derived changes (here in UV) based on presented changes of the factors affecting UV.

The text has been revised to address these points throughout.

**More specific:**

**Abstract**

I think that the included sentences such as: "All these changes are small and occur within a very variable signal." Plus the fact that is difficult to determine e.g. 1% changes with instrumentation that is just "better than 10%" have to be included here.

The abstract could be much more clear I suggest deleting the whole section starting with the new RAF text on line 19 up to line 27 and just continue with the last paragraph of new text (plus the comment above).

The Abstract has been revised in line with some of your suggestions. However, we have not included the comment above (highlighted). We agree that all these changes are small and it has been well understood that changes in total ozone, cloud cover and AOD effect the total erythema effective UV radiant exposure ($H_{er}$). These factors only contributed half of the changes in the $H_{er}$ in our study while the other half is due to other climate variables that are difficult to predict due to continuous changes in these factors from day-today and year-to-year and that climate change are also effecting these changes.  All these factors also vary depending on location, study period selected etc.

**83 UV data**

What was answered in one reviewer about not having gaps in the UV data series have to be included when describing the UV data section.
The previous reviewer response did not say that the UV dataset had no gaps but rather that the number of missing days was small (3%). This has been now added into manuscript for clarification (see Line 118-121).

It is amazing 25 years of UV data without gaps. Is the spectroradiometer calibration performed on site ?
Yes the spectroradiometer calibration is performed on site _ the spectroradiameter input optics are located close to the Chilton broadband detectors. The word "co-located" has been added in to the paragraph explaining the calibration method.

Reading the argument from lines 100-105 it shows that in the second half of the 25 year period the max drift there could have been is 20% (+10% to -10%). The uncertainty in the data for all weather conditions and across the complete time period should be provided.

Historically the Chilton head has been calibrated annually against a reference spectroradiometer which has been calibrated traceably to national standards. This principle of calibration has not changed throughout the period these data have been collected. Our consideration is that overall the uncertainties in the data are similar throughout the 25 year period, perhaps with an increase in uncertainty at the start of the dataset.

Also, after annual calibration, was data corrected if a change in the instrument was seen, and if so how? By linear interpolation back to the previous calibration, or some other method, or was the calibration only corrected going forward?

The new calibration was applied going forward and without linearly interpolating back to the previous calibration.

**107 Ozone**

Ozone data could be continuous but Dobson and Brewer monitoring schedule/data availability depends on solar zenith angle and mostly cloud conditions. So some details on these data will help the reader to understand the use of the time series.

Some extra text has been added in this section 2.2 (see Line 137-140).

**141 cloud averaging**

A use of a constant 3 hour/day averaging of cloud coverage for assessing their impact on daily solar UV changes includes the uncertainty related with the cloud changes in the remaining daytime. This should be mentioned. A more realistic solution would be to weight the cloud amount for every hour with the percentage of cloudless sky UV for the specific hour versus daily cloudless UV. However, as this requires a lot of additional work the introduced uncertainty can just be mentioned.

We believe averaging cloud cover for 3 hours/day (11-2pm) is appropriate to use here given that the UV measurements during this time contribute a large proportion of the daily $H_{er}$ overall. The text has been added to clarify this (Line 167-168)

**144 Statistics**

There is something that I cannot understand statistically. How is it possible in a time series with no gaps the Hooke results that are based on summing up all days for a year and then calculate yearly anomalies and trends to be different than averaging daily values for each month and then calculating monthly anomalies and then trends. My impression is that the sum of the monthly anomalies in a year has to match the yearly anomaly calculated by Hook for the same year (with very small differences due to the small differences of the number of days in a month).

The UV data has only 3% of missing days which have been treated here as missing values in the analysis and statistical analysis excludes them. However, in our previous analysis by Hooke et al. (2018), these missing days were filled in with the average value for each day over the entire period. There may therefore be some small differences between two analyses. We have added a text to clarify this (Line 118-121).

Trend estimate should be the same whether you consider annual mean, monthly mean, annual anomalies or monthly anomalies. However, the trend between our previous published study and here is different mainly because chosen calculation period is different for both papers. Please also see Line 573-587.

**Outliers**

More information on objective algorithms on rejecting outliers from the analysis should be provided.

We have not used any algorithms on rejecting outliers from the analysis. We did not exclude outliers neither from the dataset nor from the analysis.

The new text (lines 213-216) is contrary to the objective of the paper. If looking for underlying reasons for a trend (ozone, cloud) then data that may be particularly high/low due to ozone or cloud should not be removed from the dataset. They can be removed it if instrument problems have been identified. Otherwise the outliers seem arbitrary, and how can data be an outlier one year when it is well within the whisker value for another year?

Nowhere in the manuscript does it say that outliers were excluded in the analysis.

**Ozone and UV seasonal trends**

For the second period where clouds are almost constant and ozone plays the only role in the UV trends: the same amount of ozone changes has different effect in the UV for different seasons due to the differences in the related air masses. More specific constant ozone trends during all seasons would theoretically lead to higher UV trends in winter than in the summer months (in percent). From tables 2 and 3 seems that the opposite was found. Is there any explanation for this ?

By looking at table 2 and 3, we agree that UV trend in winter is lower than that in the summer (in %) for the second period. However, when the ozone trend values in winter (0.66% $y^{-1}$) and summer (0.13% $y^{-1}$) for the second period in Table 3 were multiplied with the RAF values in Table 4 (-1.66 and -2.18 respectively), the observed increase trend in total ozone for the second period corresponds to higher trend of -1.1% $y^{-1}$ in winter in $H_{er}$ and for summer it corresponds to the trend of -0.28%$y^{-1}$ in $H_{er}$.

**Aerosols**

The atmospheric related reasons for UV changes can be clouds, ozone, aerosol optical depth, other aerosol optical parameters, albedo, other traces gases, … (more or less ranked here based on their importance) changes. So if there are no data other than clouds and ozone you have to clarify it.
Done, see Line 181-183.

For the case of aerosols Zerefos et al., 2012 as you mention, presented changes also in non urban areas. I think that Aeronet/Chilbolton data and satellite data can be used in order to provide a hint on current speculations about a negligible 25 year effect of aerosols on this time series.

I could fast download 1 by 1 degree data around Chilton form MODIS Aqua shown below.
In addition Aeronet level 1.5 from Chilbolton station.
Modis/Aqua shows an AOD decreasing trend in the order of ~-0.003 or ~-1.5% per year. And Aeronet ~-0.007 or ~3% change per year.

This figure is a bit rough in terms of spatial resolution for Modis/Aqua and the data have been just plotted as they are with no checks at all. But some work on this aspect (e.g. lower spatial resolution Modis data or just use of the Aeronet data) can provide some more insights on the aerosol issue for the second period. Aeronet 1.5 level data also I just plotted the NASA site existing monthly mean data. They represent Chilbolton station/area.

[Figure]

If you end up on similar results then more or less results for this period agree with Zerefos et al., 2012 that state that there is a turning point that (for constant cloudiness) that the ozone increase masks the aerosol (slower rate than before) decrease for mid-latitudes.

Thank you for this useful information which is very helpful. However, the last paragraph (highlighted yellow) is not very clear to me.

We did not considered aerosols here because no ground based AOD data over the full period (1991-2015) were available near the Chilton site. However, we have now revised the manuscript a fair amount by adding new sections regarding aerosols data. We have used the AERONET dataset to analyse aerosol optical depth AOD from the Chilbolton station and comared with $H_{er}$. Please see sections 2.4, 3.5, 3.6, 4.4 and 4.5.

**Figures**

The quality of the figure 3b in the paper is not good it needs some improvement on the submitted figure format.
The fig. 3b has been redone.

I would suggest to move some of the sections describing previous works related with UV vs clouds, ozone, aerosols in the introduction section and summarize the agreement/disagreement etc findings of your work compared with the mentioned publications in the last section.
As suggested, some of the text from the discussion has moved to the introduction section (see Line 78-89).

One of the native English speakers on the author list should go through the new version of the manuscript and correct the grammar – particularly, but not limited to, the new text.
Done.